EMBO
*reports*

# c-di-GMP inhibits rRNA methylation and impairs ribosome assembly in the presence of kanamycin

Siqi Yu[1,2,3,7], Zheyao Hu [ID][4,7], Xiaoting Xu[1,3,5,7], Xiaoran Liang[1,2,3], Jiayi Shen[1,2,3], Min Liu[1,2,3], Mingxi Lin[1,2,3], Hong Chen[6], Jordi Marti [ID][4✉], Sheng-ce Tao [ID][6✉] & Zhaowei Xu [ID][1,2,3✉]

## Abstract

**Cyclic diguanosine monophosphate (c-di-GMP) is a ubiquitous bacterial secondary messenger with diverse functions. A previous *Escherichia coli* proteome microarray identified that c-di-GMP binds to the 23S rRNA methyltransferases RlmI and RlmE. Here we show that c-di-GMP inhibits RlmI activity in rRNA methylation assays, and that it modulates ribosome assembly in the presence of kanamycin. Molecular dynamics simulation and mutagenesis studies reveal that c-di-GMP binds to RlmI at residues R64, R103, G114, and K201. Structural simulations indicate that c-di-GMP quenches RlmI activity by inducing the closure of the catalytic pocket. We also show that c-di-GMP promotes antibiotic tolerance through RlmI. Binding and methylation assays indicate that the inhibitory effect of c-di-GMP on RlmI is conserved across various pathogenic bacteria. Our data suggest an unexpected role for c-di-GMP in regulating ribosome assembly under stress through the inhibition of rRNA methyltransferases.**

**Keywords** c-di-GMP; rRNA Methyltransferase; Ribosome Assembly; Antibiotic Tolerance
**Subject Category** Translation & Protein Quality

## Introduction

Cyclic diguanosine monophosphate (c-di-GMP) was first identified in *Gluconacetobacter xylinus*, where it regulates cellulose synthesis (Ross et al, 1987). Subsequent research revealed that c-di-GMP plays a crucial role in a wide range of bacterial biological processes, including motility, virulence, and host-microbe symbiosis (Hengge, 2009; Jenal et al, 2017; Obeng et al, 2023; Romling et al, 2013). In a previous study, we conducted a global screening of c-di-GMP binding proteins using an *Escherichia coli* proteome microarray, uncovering the interplay loop between c-di-GMP and protein acetylation (Xu et al, 2019). Interestingly, the microarray assay also

identified that the 23S rRNA methyltransferases RlmI and RlmE are c-di-GMP binding proteins, suggesting a functional link between c-di-GMP and ribosome assembly.

Ribosome assembly involves the processing and folding of rRNA, along with assembly with ribosomal proteins. As part of rRNA processing, rRNA methylation plays a significant role in regulating ribosome assembly. For example, the inactivation of RlmE is associated with defects in large subunit assembly (Arai et al, 2015), and RsmA, also known as KsgA, fulfills quality control requirements in the final stages of small subunit assembly (Connolly et al, 2008). Overall, there are 23 ribosomal RNA methyltransferases in *E. coli*, most of which have unresolved physiological functions. Therefore, studying the functions and regulatory factors of rRNA methyltransferase is crucial for understanding the mechanism underlying ribosome assembly.

Ribosome biogenesis is a fundamental cellular process that equips cells with molecular factories for protein production. Inhibiting ribosome assembly is considered a vital source of new drug targets (Champney, 2022; Champney, 2020). Therefore, investigating the relationship between ribosome assembly and bacterial persistence or antibiotic tolerance is essential for designing antibiotics that target ribosome assembly pathways. In gram-positive bacteria, (p)ppGpp negatively impacts ribosome assembly by inhibiting GTPase activity, thereby influencing growth and antibiotic tolerance (Corrigan et al, 2016). In addition, in Gram-negative bacteria, the effects of (p)ppGpp on antibiotic persistence primarily involve nucleotide and amino acid synthesis (Wang et al, 2020; Zhang et al, 2019). The regulatory relationship between ribosome assembly and bacterial persistence or antibiotic tolerance in Gram-negative bacteria remains poorly understood.

rRNA methylation is a significant mechanism for bacterial resistance against ribosome-targeting antibiotics. Two clinically relevant examples are 16S and 23S rRNA methyltransferases, which confer resistance by modifying conserved rRNA residues in site A or PTC, respectively. These modifications render bacteria insensitive to aminoglycosides and streptogramin B (Jeremia et al, 2023). For instance, aminoglycoside resistance in *E. coli* is conferred by the methylation of the G1405 and A1408 residues in the 16S rRNA by RsmF (Gutierrez et al, 2012) and NpmA (Wachino et al, 2007), respectively. However, the upstream regulatory factors of rRNA

[1]Key Laboratory of Gastrointestinal Cancer (Fujian Medical University), Ministry of Education, Fuzhou, China. [2]Laboratory of Scientific Research, School of Basic Medical Sciences, Fujian Medical University, Fuzhou, China. [3]Fujian Key Laboratory of Tumor Microbiology, Department of Medical Microbiology, Fujian Medical University, Fuzhou, China. [4]Department of Physics, Polytechnic University of Catalonia-Barcelona Tech, Barcelona, Catalonia, Spain. [5]Department of Endoscopy, The First Affiliated Hospital of Fujian Medical University, Fuzhou, China. [6]Shanghai Center for Systems Biomedicine, Key Laboratory of Systems Biomedicine (Ministry of Education), Shanghai Jiao Tong University, Shanghai, China. [7]These authors contributed equally: Siqi Yu, Zheyao Hu, Xiaoting Xu. ✉E-mail: jordi.marti@upc.edu; taosc@sjtu.edu.cn; xuzw@fjmu.edu.cn

methylation in the context of antibiotic pressure remain unclear, and the impact of rRNA methylation on bacterial antibiotic tolerance is not well understood.

In this study, we demonstrated that c-di-GMP binds to two 23S rRNA methyltransferases, with RlmI identified as the main effector of c-di-GMP in regulating ribosome assembly. Structural analysis revealed that c-di-GMP binds to RlmI at residues R64, R103, G114, and K201, inducing the closure of the catalytic pocket of RlmI. We further showed that c-di-GMP regulates ribosomal assembly to promote antibiotic tolerance by inhibiting RlmI activity. Finally, a sequence comparison of RlmI orthologues among bacteria indicated that some important human pathogens are conserved in the c-di-GMP-based rRNA regulatory mechanism.

# Results

## Ribosomal RNA large subunit methyltransferases are c-di-GMP effectors

In a previous study, we screened c-di-GMP-binding proteins in *E. coli* using a proteomic microarray and identified the rRNA methyltransferases RlmI and RlmE as potential c-di-GMP effectors (Xu et al, 2019) (Fig. 1A). Based on the observed binding between c-di-GMP and these methyltransferases, we hypothesized that c-di-GMP might influence rRNA methylation activity. To test this hypothesis, we assessed the activity of two methyltransferases in the presence of c-di-GMP, using rRNA methylation as our indicator. Specifically, we synthesized unmethylated 23S rRNA at positions 1932–1991 and 2522–2581 for m5C1962 by RlmI (Purta et al, 2008) and m2U2552 by RlmE (Caldas et al, 2000), respectively. These methyltransferases catalyzed the production of methylated rRNA, which was detected specific peaks in HPLC (Fig. 1B,C; Appendix Fig. S1). When c-di-GMP was introduced, methylation activity was significantly inhibited in a dose-dependent manner. When we compared the effects of additional c-di-GMP on methylation products, using the group without c-di-GMP treatment as a reference, we found that 5 μM c-di-GMP inhibited the activity of RlmI and RlmE by 49% and 31%, respectively (Fig. 1D).

## c-di-GMP inhibited ribosome assembly, with RlmI being the main effector

rRNA methylation is a prerequisite for the accurate assembly of ribosomes. We hypothesized that c-di-GMP might affect ribosomal assembly in *E. coli* (using *E. coli* BW25113 as a reference strain) by inhibiting methylation activity. To investigate the regulatory role of c-di-GMP, we constructed strains with *dgcZ* knockout and overexpression. Compared to wild-type (WT) strains, the c-di-GMP level in the *dgcZ* overexpressing strains increased by 12.2 times (Appendix Fig. S2A). We employed a sucrose density gradient (SDG) assay to detect ribosome particles, which revealed that neither the knockout nor overexpression of *dgcZ* affected the abundance of the 50S ribosomal subunit compared to the WT strain without antibiotic treatment (Fig. 2A). Since c-di-GMP as a stress response factor, we further hypothesized that the regulation of ribosome assembly by c-di-GMP might occur under antibiotic stress. We treated *E. coli* cells with kanamycin, a ribosome-targeted

antibiotic known to increase the cellular c-di-GMP level in *E. coli* by elevating dgcZ mRNA level (Ho et al, 2013; Xu et al, 2019). This increase is regulated by the RNA-binding protein CsrA (Boehm et al, 2009; Lacanna et al, 2016). Following kanamycin treatment, the c-di-GMP concentration in WT cells was 6.2-fold higher than in untreated cells (Appendix Fig. S2A). Notably, the c-di-GMP levels in *dgcZ*-defective cells did not respond to kanamycin treatment (Appendix Fig. S2A) because DgcZ functions as a synthase that mediates the kanamycin-induced increase in c-di-GMP levels. The SDG assay revealed that the ribosome disintegrated into 30S and 50S particles at low Mg²⁺ concentrations, and ~45S particles (Corrigan et al, 2016) were observed in kanamycin-treated WT cells and the *dgcZ*-defective strain complemented with a functional dgcZ gene (Δ*dgcZ::dgcZ*) (Fig. 2A). In contrast, the strain with the *dgcZ* inactivation mutation did not show the presence of 45S particles (Δ*dgcZ::dgcZ*^G206A,G207A). These results indicated that the increase in c-di-GMP levels induced by kanamycin inhibited the assembly of large ribosomal subunits in *E. coli*. In addition, we found that c-di-GMP inhibited the activity of two methyltransferases and downregulated the methylation of 23S RNA in vitro (Fig. 1D). To elucidate the role of methylation enzymes in c-di-GMP-regulated ribosome assembly, we overexpressed the two methyltransferases in kanamycin-treated WT cells. Notably, the overexpression of these methyltransferases did not affect c-di-GMP levels (Appendix Fig. S2B), but the overexpression of RlmI weakened the effect of c-di-GMP on ribosomal assembly (Fig. 2B). Thus, we conclude that c-di-GMP inhibits ribosome assembly by inactivating RlmI.

To confirm that RlmI is directly regulated by c-di-GMP and not by other c-di-GMP analogs. We employed a 100-fold excess of unlabeled c-di-GMP and its analogs (GTP, GMP, ATP, AMP, cAMP, and ppGpp) as competitive inhibitors to assess their effects on the interaction between RlmI and biotinylated c-di-GMP. The results demonstrated that unlabeled c-di-GMP effectively blocked the interaction between biotinylated c-di-GMP and RlmI, while none of the other analogs exhibited similar effects (Fig. 2C). c-di-GMP binds to its effectors via Arg residues (Chou and Galperin, 2016). To identify the binding sites on RlmI, we mutated all Arg residues to Ala in RlmI. We subsequently developed an in vitro assay in which purified RlmI mutants were incubated with biotin-c-di-GMP, subjected to UV-crosslinked, and probed with fluorescent streptavidin (Kramer et al, 2014; Shu et al, 2012). We observed that RlmI mutants with R64A and R103A exhibited a significantly weakened interaction with c-di-GMP (Fig. 2D; Appendix Fig. S3). Furthermore, when we determined the activity of RlmI mutants, both RlmI^R64A and RlmI^R103A displayed methylation activities slightly lower than that of RlmI under 5 μM rRNA substrate. Upon treatment with 20 μM c-di-GMP, the methylation activity decreased by 80%, 6%, and 32% in RlmI, RlmI^R64A, and RlmI^R103A, respectively (Fig. 2E). The diminished effect of c-di-GMP on RlmI in the R64A and R103A mutant strains, compared to the WT strain, confirmed that R64 and R103 are the key sites involved in c-di-GMP binding to RlmI.

To validate whether RlmI is the primary effector of c-di-GMP in ribosome assembly, we eliminated the effect of c-di-GMP on RlmI by mutating the binding sites R64A and R103A. Under kanamycin treatment, we analyzed WT, *rlmI*^R64A, *rlmI*^R103A, and Δ*rlmI* strains using the sucrose density gradient (SDG) assay. We examined the

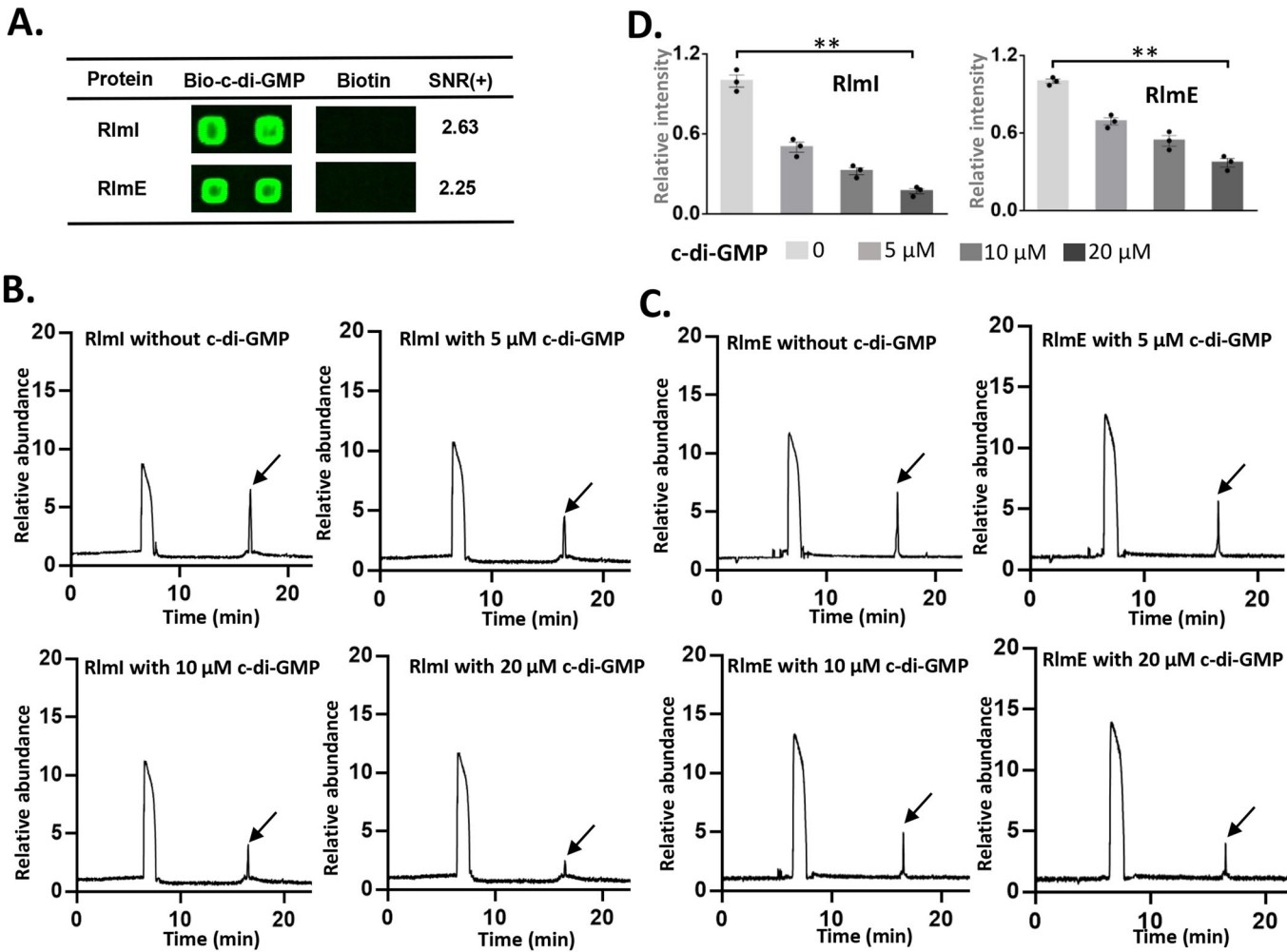

**Figure 1. Ribosomal RNA large subunit methyltransferases are c-di-GMP effectors.**

(A) *E. coli* proteome microarrays were probed with biotin-c-di-GMP and biotin. Obvious differences in the binding of RlmI and RlmE on the microarrays incubated with biotin-c-di-GMP and biotin were observed. Two spots per protein were observed, and the positive signal-to-noise ratio [(SNR) (+)] represented the average SNR of the two duplicate spots. (B, C) In vitro methylation reaction. The synthesized rRNA fragments were used for in vitro methylation enzyme activity testing. The HPLC peaks are derived from RlmI (B) and RlmE (C) after treatment with 0, 5, 10, and 20 μM c-di-GMP, respectively. The second peak (black arrow) represents the methylated rRNA, which is used to calculate the activity of the methyltransferase. (D) Quantitative results of the HPLC peak. The methylated rRNAs, indicated by the second peak, were detected by HPLC and quantified by area under the curve. The bar chart shows the relative enzyme activity with the data points, using the reaction without the addition of c-di-GMP as the baseline ($n = 3$ biological replicates, mean ± s.e.m.; **$p < 0.01$ ($p = 0.0011$ between 0 μM c-di-GMP and 20 μM c-di-GMP in RlmI; $p = 0.00059$ between 0 μM c-di-GMP and 20 μM c-di-GMP in RlmE), two-tailed Student's t-test). Source data are available online for this figure.

impact of c-di-GMP on ribosome biogenesis by performing SDG to analyze changes in the 70S and 50S ribosomal populations under 10 mM MgCl₂ conditions. When c-di-GMP levels were elevated, we observed a significant accumulation of 50S subunits. However, this accumulation was markedly reduced when mutations were introduced at the R64 or R103 sites of RlmI, indicating that c-di-GMP acts through RlmI to influence the formation of intact 70S ribosomes (Fig. 3A). In addition, we compared methylation levels of C1962 in the 50S and 70S ribosomes under high c-di-GMP expression and 10 mM MgCl₂ conditions. We found that the 50S subunits displayed significantly lower levels of m⁵C1962 methylation compared to the 70S particles, suggesting that proper methylation at C1962 is associated with the maturation of the ribosomal large subunit (Fig. 3B).

To more accurately observe ribosome assembly intermediates, we performed density gradient centrifugation under 0.5 mM MgCl₂ conditions. The results demonstrated that under high c-di-GMP expression, there was a significant accumulation of 45S ribosomal intermediates. However, when the RlmI R64A or R103A mutations were present, the accumulation of 45S ribosomes was significantly reduced (Fig. 3A). Furthermore, we observed that in the context of high c-di-GMP expression, the 45S ribosomes exhibited significantly lower levels of m⁵C1962 methylation compared to the 50S subunits. These findings suggest that methylation at C1962 is correlated with the proper assembly and maturation of the ribosomal large subunit (Fig. 3B). Taken together, these results demonstrate that RlmI plays a crucial role in ribosomal assembly under kanamycin stress and that c-di-GMP regulates RlmI's activity.

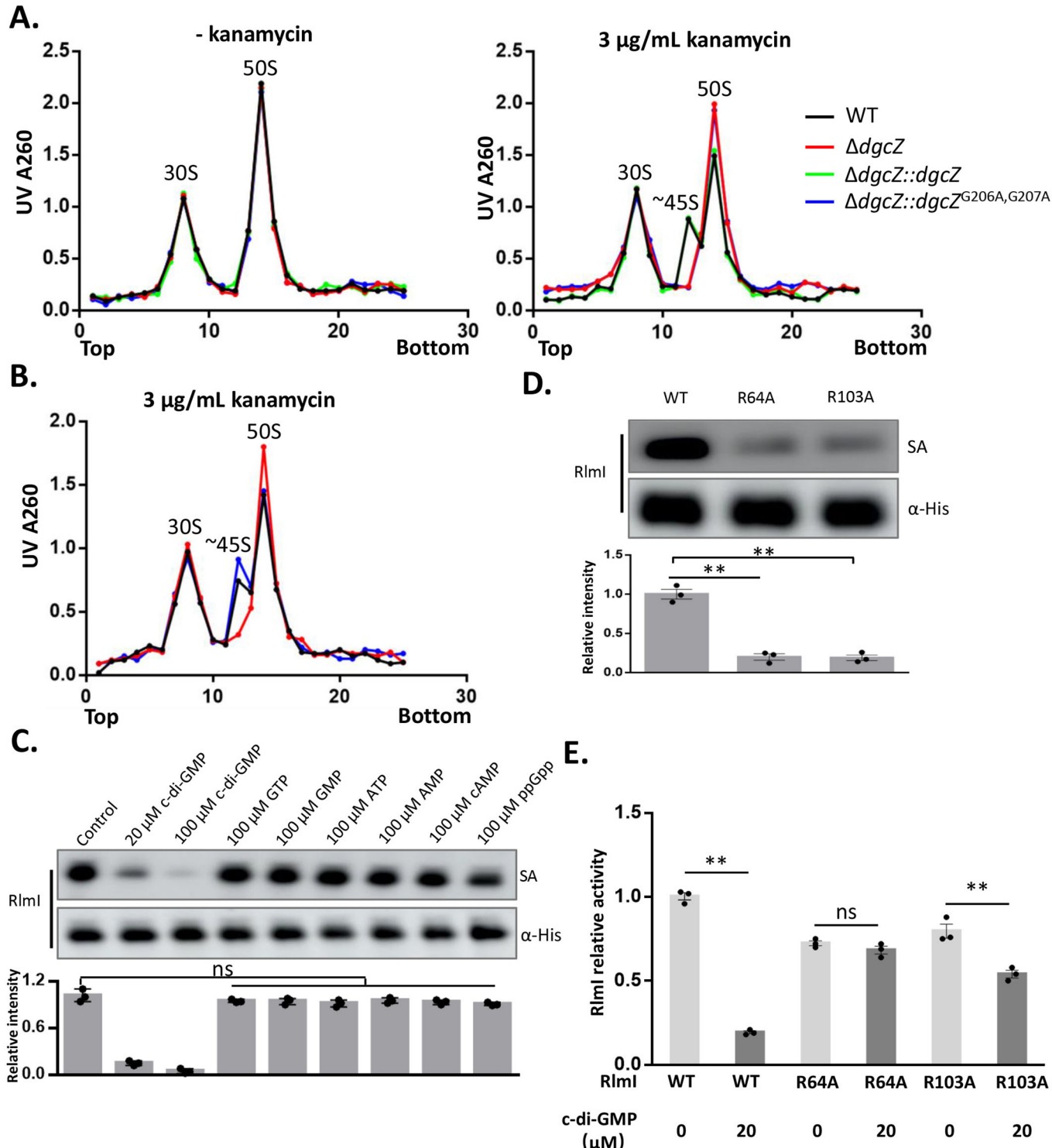

## c-di-GMP induces the closure of the catalytic pocket of RlmI

To elucidate the structural mechanism by which c-di-GMP regulates RlmI enzyme activity, we investigated the conformational changes of RlmI during its interaction with c-di-GMP in an aqueous ionic solution using molecular dynamics (MD)

simulations. The root mean square deviation (RMSD) indicated the fluctuations and stability of the conformations of RlmI, while the root mean square fluctuation (RMSF) revealed flexibility throughout the simulation period. Analysis showed that residues 160–170, 302–320, and 370–390 were mainly involved in the conformational fluctuations of RlmI (Fig. 4A). We labeled residues 160–170 as "Domain-I" (DM-I), residues 302–320 as "Domain-II"

**Figure 2. c-di-GMP inhibits ribosome assembly, with RlmI as the main effector.**

(A) SDG assay for the strains with elevated c-di-GMP. c-di-GMP was elevated by treatment with kanamycin or overexpression of DgcZ, and the ribosome particles were assayed by SDG. The corresponding three peaks represent the ribosome particles of 30S, pre-50S, and 50S. (B) SDG assay for the strains overexpressing two methyltransferases. RlmI and RlmE were overexpressed under kanamycin treatment, and the ribosome particles were assayed by SDG. (C) c-di-GMP analog competitive assay. Streptavidin represents the interaction signals, and α-His represents the protein levels. The bar chart shows the relative intensity of streptavidin with the data points ($n = 3$ biological replicates, mean ± s.e.m.; ns: no significant difference, two-tailed Student's t-test). (D) The arginine on RlmI was mutated to alanine, and the interaction of c-di-GMP and RlmI mutants was determined. The results indicated that R64A and R103A weakened the binding of c-di-GMP and RlmI. Streptavidin represents the interaction signals, and α-His represents the protein levels. The bar chart shows the relative intensity of streptavidin with the data points ($n = 3$ biological replicates, mean ± s.e.m.; **$p < 0.01$ ($p = 0.0016$ between WT and R64A; $p = 0.0011$ between WT and R103A), two-tailed Student's t-test). (E) In vitro methylation assay of the two RlmI mutants. The synthesized rRNA fragments were used as substrates, and the reaction products were analyzed by HPLC. The bar chart shows the relative activity of RlmI with the data points ($n = 3$ biological replicates, mean ± s.e.m.; ns: no significant difference, **$p < 0.01$ ($p = 0.00046$ between 0 μM c-di-GMP and 20 μM c-di-GMP in WT; $p = 0.057$ between 0 μM c-di-GMP and 20 μM c-di-GMP in R64A; $p = 0.0082$ between 0 μM c-di-GMP and 20 μM c-di-GMP in R103A), two-tailed Student's t-test). Source data are available online for this figure.

(DM-II), and residues 370–390 as "Domain-III" (DM-III). DM-I and DM-III are the regions of the protein that correspond to the RNA-binding area, whereas DM-II is located near the S-adenosyl-L-methionine (AdoMet) binding-related area. An overall view of the evolution of RlmI fluctuations revealed a distinct conformational fluctuation of approximately 0.8 μs during the simulation (average of Trajectory #1 and Trajectory #2) (Fig. 4B). Combining RMSD, RMSF, and trajectory analysis results, we identified two states for RlmI during its interaction with c-di-GMP: State-I and State-II. c-di-GMP interacts with the RNA-binding area (State-I), and the domain DM-III shut down after 0.8 μs. The results suggested that (1) c-di-GMP can interact with the RNA-binding domains and then induce the closure of DM-III, and that (2) the "on-off" of the RNA-binding area was mainly embodied by DM-III to a large extent (Fig. 4C and Movie EV1–2). The dynamic process of c-di-GMP-induced conformational rearrangements in the active domain of RlmI was similar to that of other c-di-GMP effectors such as YcgR (Hou et al, 2020), FleQ (Matsuyama et al, 2015), and CheR1 (Yan et al, 2018).

### Interactions of R64, R103, G114, and K201 residues of RlmI with bound c-di-GMP

We employed Gibbs free energy analysis to identify the dominant conformation of RlmI and c-di-GMP complex via molecular dynamics simulations. The Gibbs free energy surfaces for the two runs and their average values are shown (Fig. 5A), using RMSD and radius of gyration used as the variables. We identified the free energy basin, the one with the lowest free energy (set to 0 kJ/mol) (Fig. 5A, yellow point), and found that the corresponding regions were almost overlapping for the three sets (Fig. 5B). Thus, the results indicated the two independent simulated trajectories as convergent and physically equivalent.

To explore the binding sites of c-di-GMP and RlmI, we superimposed the stable-state configurations of RlmI and c-di-GMP for the three sets (Fig. 5B). Two independent trajectories, #1 and #2, were taken into consideration for the computational analysis and the average was selected for convergence and physical equivalence analysis. The average conformation showed that R64 and G114 together stabilize the guanosine moiety of c-di-GMP. Correspondingly, R103 and K201 act to stabilize the negatively charged region of c-di-GMP. It is evident that R103 and K201 formed a stable hydrogen bond with the oxygen atom of the phosphate group of c-di-GMP.

Noncovalent interactions, including hydrogen bonds, coordination bonds, and salt bridges, are crucial for maintaining the tertiary structure of proteins. The all-atom-level precision of the molecular dynamics simulations, we analyzed the hydrogen bond interaction map of c-di-GMP with RlmI using time-dependent atomic site distances between selected atomic sites to uncover the interaction mode of c-di-GMP with RlmI, providing guidance for further experimental verification. Atomic detail sketches of c-di-GMP and the main residues described in this section are provided. While labeling the amino acid residues in the hydrogen bond interaction map of c-di-GMP with RlmI, we also labeled the lifetime of hydrogen-bonding interactions between c-di-GMP and the corresponding amino acid residues. Considering that our molecular dynamics simulation spanned a timeframe of 2 μs, we subsequently performed site mutation verification on residues with hydrogen bond interaction lifetimes exceeding 400 ns. Six amino acid residues from RlmI were selected as the potential binding sites for c-di-GMP: R64, R103, E108, G114, T116, and K201 (Fig. 5C). Atom-atom distances as a function of time and bond lifetimes are presented in Appendix Figs. S5–15.

We employed the streptavidin blotting assays to determine the interaction between c-di-GMP and RlmI mutants, aiming to validate the results obtained from the molecular dynamics simulations. The results revealed that the amino acid residues R64, R103, G114, and K201 were crucial for the binding of c-di-GMP to RlmI. In addition, E108A and T116A of RlmI slightly affected c-di-GMP binding (Fig. 5D). We next performed isothermal titration calorimetry (ITC) titrations with these mutants and determined $K_d$ values of 1.3, 102.3, 76.5, 148.6, and 401.2 μM for RlmI, RlmI$^{R64A}$, RlmI$^{R103A}$, RlmI$^{G114A}$, and RlmI$^{K201A}$, respectively (Fig. 5E; Appendix Fig. S16). We found that the stoichiometries of RlmI and its variants with c-di-GMP are not significantly different, each showing a 1:1 binding ratio (Fig. 5E). Furthermore, we observed that the activity of RlmI$^{K201A}$ did not significantly differ between the 20 μM c-di-GMP treatment group and the c-di-GMP free group (Fig. 5F). The results suggested that the R64, R103, G114, and K201 residues of RlmI were the critical sites for c-di-GMP binding.

### c-di-GMP regulates RlmI to promote antibiotic tolerance

Given that a close correlation exists between c-di-GMP, ribosomal assembly, and antibiotic resistance (Gomez et al, 2017; Gupta et al, 2014), we hypothesized that c-di-GMP regulates ribosomal

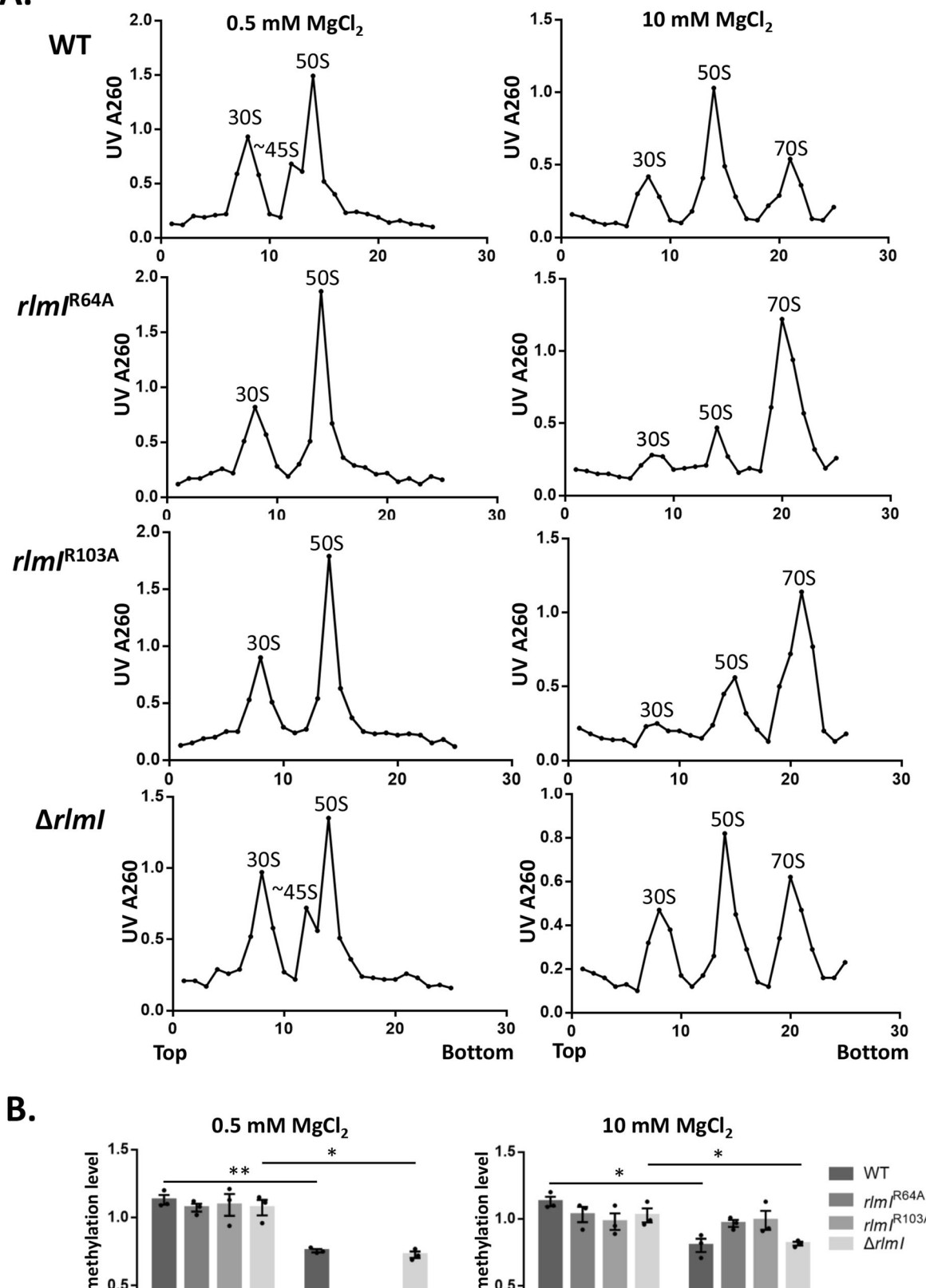

◀ **Figure 3.   Ribosome assembly and C1962 methylation in response to varying Mg²⁺ concentrations and c-di-GMP levels in the presence of kanamycin.**

(A) Sucrose density gradient (SDG) analysis of ribosome profiles in RlmI-depleted and RlmI-mutant strains treated with kanamycin. Ribosomal particles were separated by SDG under two different MgCl$_2$ concentrations (0.5 mM and 10 mM) to evaluate changes in ribosome assembly and stability. Comparison of the resulting profiles highlights how varying MgCl$_2$ conditions influence ribosome distribution and reveals the effects of RlmI depletion or mutation on ribosome integrity and function ($n = 3$ biological replicates). (B) Methylation levels of C1962 in 23S rRNA were examined under varying MgCl$_2$ conditions to assess how assembly states affect ribosomal modification. At 10 mM MgCl$_2$, both the 50S subunits and fully assembled 70S ribosomes were analyzed, while at 0.5 mM MgCl$_2$, the 50S and partially assembled 45S components were examined. This comparison highlights the influence of MgCl$_2$ concentration and ribosomal assembly state on the C1962 methylation status within the ribosome ($n = 3$ biological replicates, mean ± s.e.m.; **$p < 0.01$, *$p < 0.05$; (In the condition of 0.5 mM MgCl$_2$, $p = 0.0039$ between 50S and 45S in WT, $p = 0.048$ between 50S and 45S in △$rlmI$; In the condition of 10 mM MgCl$_2$, $p = 0.026$ between 50S and 70S in WT, $p = 0.034$ between 50S and 70S in △$rlmI$), two-tailed Student's t-test). Source data are available online for this figure.

assembly to promote bacterial persistence or antibiotic tolerance by inhibiting RlmI activity. To test this, we interfered with the interaction of c-di-GMP and RlmI by introducing the K201 mutation in endogenous RlmI ($rlmI^{K201A}$), $dgcZ$ depletion (△$dgcZ$), $dgcZ$ overexpression (△$dgcZ$ $dgcZ^+$) and both mutants (△$dgcZ$ $rlmI^{K201A}$). We subsequently determined the c-di-GMP level and methylation level of 23S rRNA C1962. The results indicated no significant changes in c-di-GMP levels and C1962 methylation levels across WT, △$dgcZ$, $rlmI^{K201A}$, and △$dgcZ$ $rlmI^{K201A}$ without kanamycin treatment (Fig. 6A,B). However, with kanamycin treatment, the c-di-GMP levels of WT and $rlmI^{K201A}$ increased approximately six times compared with △$dgcZ$ and △$dgcZ$ $rlmI^{K201A}$. The c-di-GMP levels of △$dgcZ$ $dgcZ^+$ increased approximately eleven times compared with those of WT (Fig. 6A). Methylation analysis showed that elevated c-di-GMP levels (WT, △$dgcZ$ $dgcZ^+$) significantly decreased C1962 methylation, while the K210A mutation ($rlmI^{K201A}$) abolished the effect of c-di-GMP (Fig. 6B). These findings provide in vivo evidence that c-di-GMP regulates 23S rRNA methylation through RlmI.

To investigate the role of c-di-GMP regulation of RlmI in bacterial antibiotic tolerance, we examined wild-type (WT), $rlmI^{K201A}$, and △$dgcZ$ strains. The △$dgcZ$ strain, in which the major c-di-GMP synthase DgcZ is deleted in *E. coli*, exhibits a reduced intracellular c-di-GMP level, while the $rlmI^{K201A}$ strain carries a mutation at the K201 site of RlmI, abolished the regulation of c-di-GMP on RlmI. To assess antibiotic sensitivity, bacterial killing curves were generated for the three strains in the presence of kanamycin and ampicillin at 20× and 100× MIC concentrations. The results revealed that both the $rlmI^{K201A}$ and △$dgcZ$ strains exhibited increased sensitivity to antibiotics compared to the WT strain, suggesting that both c-di-GMP and RlmI contribute to modulating antibiotic response (Fig. 6C).

Notably, the MDK$_{99}$ for the $rlmI^{K201A}$ strain was significantly lower than that of the WT strain at both 20× and 100× kanamycin concentrations, measuring 104.79 and 87.25 min, respectively. Similarly, the MDK$_{99.99}$ for $rlmI^{K201A}$ was markedly reduced to 196.62 and 189.40 min under the same kanamycin concentrations compared to the WT. These reductions were also observed when treated with ampicillin. These results indicate that, compared to the WT, the $rlmI^{K201A}$ strain exhibits significantly decreased antibiotic tolerance. In addition, the △$dgcZ$ strain also displayed significantly reduced MDK$_{99}$ and MDK$_{99.99}$ values compared to the WT, indicating that c-di-GMP modulates antibiotic sensitivity (Fig. 6D). Based on this evidence, we propose that c-di-GMP promotes bacterial tolerance to antibiotics through its regulation of RlmI.

## The effect of c-di-GMP on RlmI may be conserved in multiple pathogenic bacteria

c-di-GMP is a ubiquitous bacterial secondary messenger, and RlmI is highly conserved in bacteria. Thus, we hypothesized that the binding and inhibition of c-di-GMP with RlmI from *E. coli* was the same for the RlmI homologs in other bacteria. To test this hypothesis, we aligned RlmI protein sequences from a series of highly diverse bacteria and found that the c-di-GMP binding region well conserved in these bacteria (Fig. 7A). Then, we selected *Salmonella typhimurium*, *Klebsiella pneumoniae*, and *Vibrio cholerae* as the exemplary members of this conserved set. Our analysis revealed that RlmI$^{S. typhimurium}$, RlmI$^{K. pneumoniae}$, and RlmI$^{V. cholerae}$ could bind to c-di-GMP, with binding abolished upon mutation of the lysine in RlmI (Fig. 7B). Moreover, the in vitro activity analysis showed that similar to RlmI$^{E.coli}$, the aforementioned three RlmI homologs exhibited methylase activity for 23S rRNA and this activity could be inhibited by c-di-GMP (Fig. 7C). Thus, the effect of c-di-GMP on RlmI may be conserved in multiple pathogenic bacteria.

## Discussion

c-di-GMP is a crucial secondary messenger in prokaryotes, and rRNA methylation occurs in both prokaryotes and eukaryotes. This study revealed that c-di-GMP binds to two rRNA methyltransferases, inhibiting their activities, with RlmI identified as the primary effector of c-di-GMP in ribosome assembly. Molecular dynamics simulations revealed the binding sites and models of c-di-GMP interacting with RlmI. In addition, killing assays demonstrated that c-di-GMP inhibits ribosome assembly, thereby promoting antibiotic tolerance in *E. coli*. This research establishes a regulatory pathway linking c-di-GMP to ribosomal functions, underscoring the role of c-di-GMP in antibiotic tolerance.

Previous studies have reported that c-di-GMP regulates mature ribosome function through RimK in *Pseudomonas* (Grenga et al, 2020; Little et al, 2016), EF-P in *Acinetobacter baumannii* (Guo et al, 2022), and Vc2 riboswitches in *V. cholerae* (Inuzuka et al, 2018). c-di-GMP regulates the glutamate ligase RimK, which catalyzes glutamate residues to the C-terminus of the ribosomal protein RpsF to affect ribosomal function (Grenga et al, 2020; Little et al, 2016). The binding of c-di-GMP enhances the function of EF-P, promoting translation efficiency and modulating bacterial physiology and virulence (Guo et al, 2022). In addition, c-di-GMP binds to the Vc2 riboswitch, inducing structural changes that result in switch-OFF and switch-ON states of translational initiation (Inuzuka et al, 2018). This study revealed that the role of c-di-GMP affects ribosome assembly, offering

                                                                      

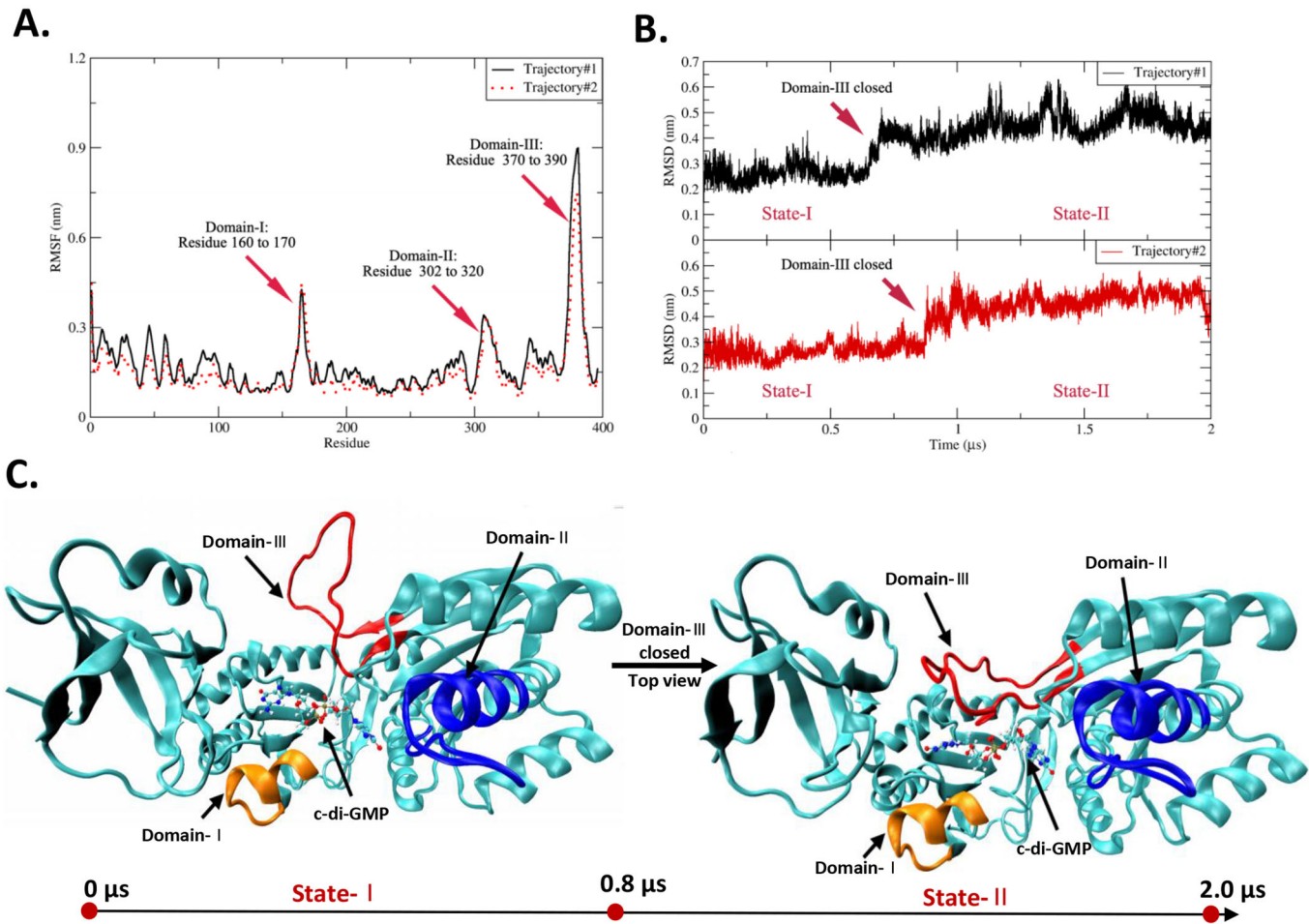

**Figure 4. The simulated interaction model of c-di-GMP and RlmI.**

The interaction model of c-di-GMP with RlmI was simulated using the CHARMM-GUI platform. (A) RMSD value of RlmI during interaction with c-di-GMPs. The arrow marks the main changes at the residue level. The solid and dashed lines represent two simulated trajectories, and the degree of overlap represents the convergence of two trajectories. (B) RMSF value of RlmI during interaction with c-di-GMP. The arrow indicates the main changes on the timeline. (C) Representative snapshots of the transition between RlmI State-I and State-II. The arrow marks the main changes in DM-III. Source data are available online for this figure.

a new perspective on c-di-GMP in ribosome regulation. Numerous accessory factors play a role in guiding the ribosome assembly process, including GTPases, rRNA modification enzymes, helicases, and maturation factors (Davis and Williamson, 2017). Our findings establish c-di-GMP as an upstream regulatory signal for rRNA modification, creating a connection between environmental stimuli and ribosome function.

RlmI is a large ribosomal RNA subunit methyltransferase that specifically methylates cytosine at position 1962 (m5C1962) of 23S rRNA. Previous studies indicated that RlmI depletion did not lead to abnormal ribosome assembly or growth arrest of *E. coli* at 20 °C or 37 °C (Pletnev et al, 2020). Indeed, we found that RlmI depletion did not affect the abundance of 50S ribosome subunits compared with WT strains in the absence of antibiotics. However, upon kanamycin treatment, ~45S particles were observed in the △*rlmI* cells. Thus, RlmI plays a key role in ribosomal assembly under kanamycin stress. As deletion of most ribosomal methyltransferases does not cause significant phenotypic changes, these studies have demonstrated that the function of methylases under different

growth conditions may help understand the physiological significance of ribosome assembly. In addition, we have not yet addressed the specific role of kanamycin in ribosome assembly, a question that remains of significant interest. Understanding how kanamycin interacts with the ribosomal assembly process is a key area of curiosity for us. To investigate this, identifying the components of 45S ribosomal intermediates and analyzing their structures will be crucial. This approach will provide valuable insights into the molecular mechanisms underlying the influence of kanamycin on ribosome assembly and its potential effects on bacterial protein synthesis.

RlmE plays a critical role in ribosome assembly by modifying the methylation status of m²U2552, thereby influencing the structural and functional integrity of the ribosome. Our study reveals that the secondary messenger c-di-GMP can bind to RlmE, and in vitro assays demonstrate that c-di-GMP effectively inhibits the methyltransferase activity of RlmE. However, in vivo experiments indicate that the restoration of RlmE expression does not alter the regulatory effect of c-di-GMP on ribosome assembly. This discrepancy suggests

 

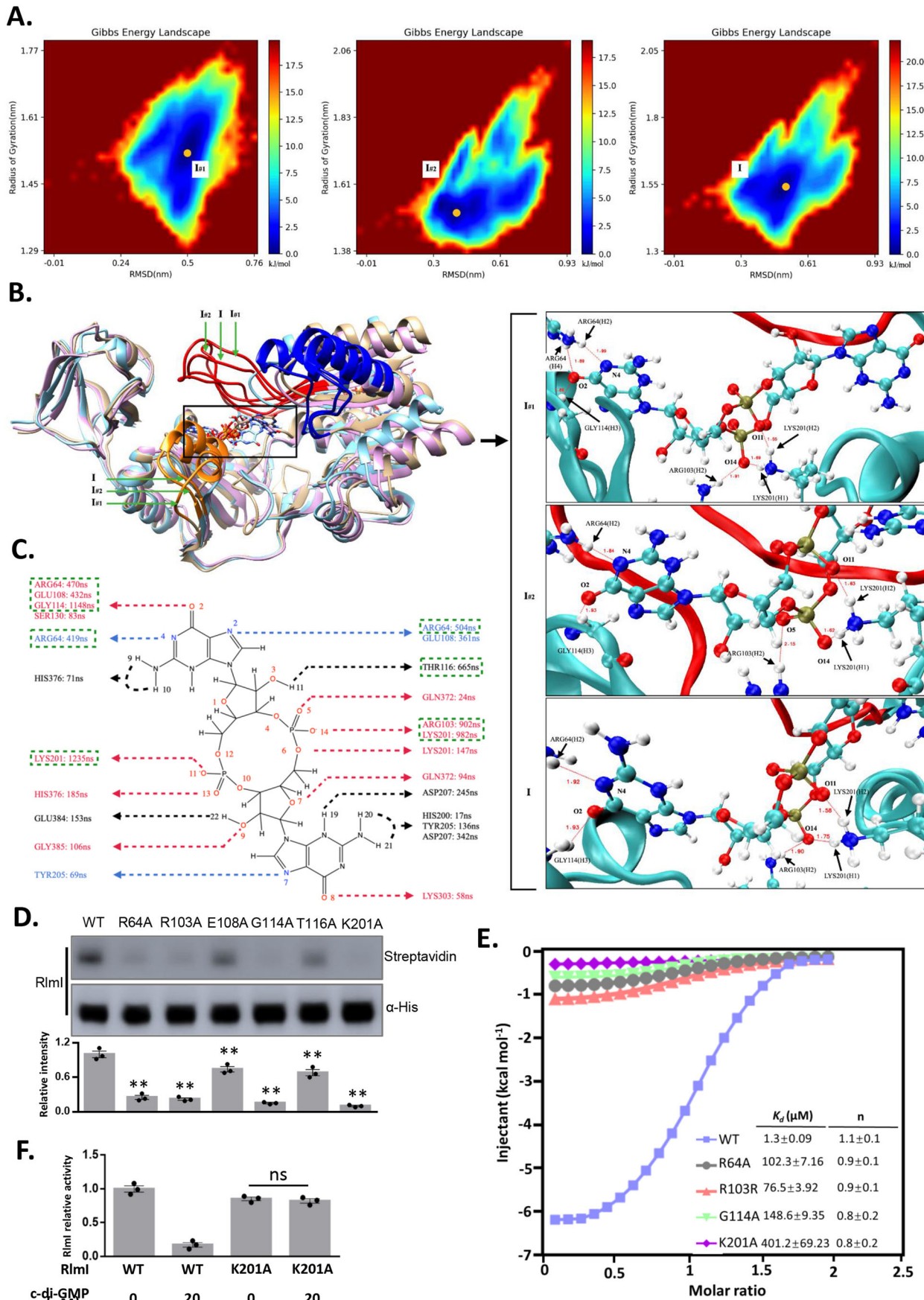

**Figure 5. Determination of the binding sites of c-di-GMP on RlmI.**

(A) Gibbs free energy landscapes illustrating the binding of c-di-GMP to RlmI in MD. I#1, I#2, and I represent trajectories 1, 2, and average, respectively. (B) The interaction model of c-di-GMP and RlmI in MD. The key binding sites R64, R103, G114, and K201 are highlighted. I#1, I#2, and I represent trajectories 1, 2, and average, respectively. (C) Simulated interaction sites of c-di-GMP and RlmI. The interaction sites between c-di-GMP and RlmI in MD simulation are marked in the plane diagram of c-di-GMP. The annotated duration represents the interaction time between c-di-GMP and RlmI in the 10 μs MD simulation, with green boxes highlighting residues with interaction times greater than 400 ns. (D) Streptavidin blotting assays comparing the WT and mutant RlmI. The potential binding sites within RlmI were mutated to alanine, and the interaction with c-di-GMP was determined. Streptavidin represents the interaction signals, and α-His represents the protein levels. The intensity of WT was used as the reference for statistical analysis of the variants. The bar chart shows the relative intensity of streptavidin with the data points ($n = 3$ biological replicates, mean ± s.e.m.; **$p < 0.01$ ($p = 0.00077$ between WT and R64A; $p = 0.0021$ between WT and R103A; $p = 0.0020$ between WT and E108A; $p = 0.0029$ between WT and G114A; $p = 0.00033$ between WT and T116A; $p = 0.0026$ between WT and K201A), two-tailed Student's t-test). (E) ITC analysis of the binding between c-di-GMP and RlmI. c-di-GMP was titrated into RlmI and its mutants in triplicate. (F) In vitro methylation assay of the RlmI mutants. The bar chart shows the relative activity of RlmI with the data points ($n = 3$ biological replicates, mean ± s.e.m.; ns: no significant difference, two-tailed Student's t-test). Source data are available online for this figure.

that while c-di-GMP interacts with RlmE, its primary target may lie elsewhere. Notably, c-di-GMP exhibits a more potent inhibitory effect on RlmI compared to RlmE, implying that RlmI could be the main receptor mediating c-di-GMP's regulatory functions, with RlmE serving as a secondary receptor. Furthermore, the precise mechanisms by which RlmE regulate ribosome assembly through rRNA methylation in vivo remain to be elucidated. As research progresses to further elucidate the role of RlmE in ribosome assembly, we may uncover additional layers of regulation and better understand the physiological significance of c-di-GMP's modulation of RlmE activity. This could provide deeper insights into the intricate network of ribosome biogenesis and its regulation by second messengers like c-di-GMP.

Our findings suggest a significant link between rRNA methylation and antibiotic tolerance. Specifically, the inhibition of rRNA methylation by c-di-GMP appears to enhance antibiotic tolerance in *E. coli*, indicating that disruptions in this fundamental process can lead to survival advantages under antibiotic pressure. This relationship underscores the importance of ribosomal integrity in maintaining cellular function and resisting external stresses. Moreover, the regulation of rRNA methylation through molecules like c-di-GMP highlights a sophisticated mechanism by which bacteria can modulate their responses to antibiotic exposure. Understanding these connections could lead to novel strategies for combating antibiotic tolerance by targeting ribosome assembly pathways, thereby restoring the efficacy of existing antibiotics. Further investigations are needed to elucidate the precise molecular mechanisms involved and their implications for bacterial adaptation and resistance.

In summary, we identified rRNA methyltransferases as the novel c-di-GMP effectors through *E. coli* proteome microarray assay. The functional analysis revealed an unexpected role of c-di-GMP in regulating ribosome assembly by inhibiting rRNA methylases, highlighting the physiological function of the regulatory axis in antibiotic tolerance.

## Methods

### Reagents and tools table

| Reagent/resource | Reference or source | Identifier or catalog number |
|---|---|---|
| **Experimental models** | | |
| *E. coli* BW25113 | Poteete et al, 2001 | |

| Reagent/resource | Reference or source | Identifier or catalog number |
|---|---|---|
| **Recombinant DNA** | | |
| pCA24N | This study | |
| pGEX4T-1 | This study | |
| pET28a | This study | |
| **Oligonucleotides** | | |
| *rlmI* primer | This study | Sangon oligonucleotide custom order |
| *rlmE* primer | This study | Sangon oligonucleotide custom order |
| *gapA* primer | This study | Sangon oligonucleotide custom order |
| C1962 primer | This study | Sangon oligonucleotide custom order |
| 23S rRNA primer | This study | Sangon oligonucleotide custom order |
| **Antibodies** | | |
| Anti-m5C antibody | CST | 28692 |
| IRDye 800CW Conjugated Streptavidin | LI-COR Biosciences | 926-32230 |
| Anti-His antibody | Millipore | 05-949 |
| **Chemicals, enzymes, and other reagents** | | |
| QuikChange Site-Directed Mutagenesis Kit | Agilent Technologies | 200518 |
| $MgSO_4 \cdot 7H_2O$ | Sangon | A610329 |
| $K_2HPO_4$ | Sangon | A610447 |
| $C_6H_8O_7 \cdot H_2O$ | Sangon | A502123 |
| $NaNH_4HPO_4 \cdot 4H_2O$ | Sangon | A502816 |
| $MgCl_2 \cdot 6H_2O$ | Sangon | A610328 |
| $NH_4Cl$ | Sangon | A417703 |
| KCl | Sangon | A610440 |
| Acetate | Sangon | A428556 |
| Kanamycin | Sangon | A600286 |
| Ampicillin | Sangon | A430258 |
| Tryptone | Sangon | A650217 |
| Yeast extract | Sangon | A610961 |

| Reagent/resource | Reference or source | Identifier or catalog number |
|---|---|---|
| NaCl | Sangon | A610476 |
| Glycerol | Sangon | A450039 |
| c-di-GMP | Biolog | C057 |
| Biotin-c-di-GMP | Biolog | B098 |
| GTP | Sigma-Aldrich | G8877 |
| GMP | Sigma-Aldrich | G7504 |
| ATP | Sigma-Aldrich | A6559 |
| AMP | Sigma-Aldrich | A2252 |
| cAMP | Sigma-Aldrich | A9501 |
| cGMP | Sigma-Aldrich | G7504 |
| ppGpp | Jena bioscience | NU-885S |
| HEPES | Sigma-Aldrich | H0527 |
| AdoMet | Sigma-Aldrich | A4377 |
| Protector RNase inhibitor | Sigma-Aldrich | R7397 |
| Protein A/G beads | Thermo Fisher Scientific | |
| Glycine | Sangon | A610236 |
| RNA extraction kit | Sangon Biotech | |
| Reverse transcription kit Real-time PCR kit | Applied Biosystems | 11736051 |
| Lysozyme | Sigma-Aldrich | L2879 |
| Benzonase | Sigma-Aldrich | E8263 |
| **Software** | | |
| R | www.r-project.org | |
| GraphPad Prism | www.graphpad.com | |
| CHARMM-GUI | www.charmm-gui.org | |
| UCSF Chimera | www.cgl.ucsf.edu/chimera | |
| VMD | www.ks.uiuc.edu/Research/vmd/ | |
| Origin 7.0 | OriginLab | |
| **Other** | | |
| MicroCal iTC200 system | GE Healthcare | |
| Odyssey Infrared Imaging System | LI-COR Biosciences | |
| UPLC-IM-MS | Waters Corporation | |

## E. coli strains and plasmids

E. coli BW25113 was used as the reference strain in this study, and plasmids pCA24N and pGEX4T-1 were utilized to overexpress methyltransferases and their mutants. In addition, pET28a was employed for RlmI of S. typhimurium, K. pneumoniae, and V. cholerae. Recombinant RlmI mutations were performed using a QuikChange Site-Directed Mutagenesis Kit (#200518, Agilent Technologies, USA).

The strains described earlier were cultivated in Vogel-Bonner medium (0.81 mM MgSO$_4$·7H$_2$O, 43.8 mM K$_2$HPO$_4$, 10 mM C$_6$H$_8$O$_7$·H$_2$O, and 16.7 mM NaNH$_4$HPO$_4$·4H$_2$O) with 10 mM acetate at 25 °C for functional analysis. For kanamycin treatment, 3 μg/mL kanamycin was added to the Vogel-Bonner medium.

For protein purification, strains were grown in lysogeny broth (10 g tryptone, 5 g yeast extract, 10 g NaCl per 1 L) medium (LB medium), and induced with 0.2 mM IPTG at 22 °C for 12 h.

## Construction of endogenous DgcZ and RlmI mutants in E. coli

DgcZ and RlmI mutants were constructed using the Red-recombination system based on the E. coli BW25113 strain (Poteete, 2001), as previously described (Tu et al, 2015). For kanamycin-resistant E. coli strains, rlmI$^{K201A}$ mutants were constructed using the Red-recombination system with ampicillin as the screening antibiotic.

## In vitro methyltransferase activity assay

In vitro methylation assays were conducted to investigate the methylation activity of recombinant enzymes RlmI and RlmE on rRNA substrates in the presence of c-di-GMP. The recombinant RlmI and RlmE proteins were first purified and dialyzed into PBS (phosphate-buffered saline) containing 20% glycerol. The dialyzed proteins were flash-frozen in liquid nitrogen and stored at −80 °C until use. For the assays, the purified recombinant enzymes were resuspended in a reaction buffer, and the final concentration of each enzyme was 3 μM. c-di-GMP was added at varying concentrations (5 μM, 10 μM, or 20 μM) to assess its dose-dependent effect on methylation. The reaction buffer contained 40 mM HEPES (pH 7.6), 100 mM NH$_4$Cl, 10 mM MgCl$_2$, and 1 mM AdoMet (S-adenosylmethionine), which served as the methyl donor.

The reaction mixture was incubated at 37 °C for 30 min to allow sufficient interaction between the enzymes and c-di-GMP. After the initial incubation, 5 μM rRNA substrates (which had been pre-prepared) were added to the reaction, and the incubation continued at 37 °C for an additional 90 min to facilitate methylation. To ensure that the enzymes' methylation activity was specifically directed toward the rRNA substrates, reactions were carefully monitored for appropriate timepoints.

To terminate the reaction, the samples were heated at 95 °C for 10 min, which effectively denatured the enzymes and stopped further methylation. The resulting products were then subjected to centrifugation at 10,000 × g for 10 min at 4 °C to remove any insoluble material or unreacted components. The supernatant containing the methylated rRNA substrates was carefully collected for downstream analysis.

The samples were analyzed using high-performance liquid chromatography (HPLC). Briefly, the samples were injected into a C-18 column (Alltima C18 4.6 250 mm$^2$) and analyzed via reversed-phase HPLC (Shimadzu, Japan). Solution A [0.065% trifluoroacetic acid in 100% water (v/v)] and B [0.05% trifluoroacetic acid in 100% acetonitrile (v/v)] were used in a gradient program (0.01 min with 5% Solution B, 20 min with 65% Solution B, 20.01 min with 95% Solution B, 31 min with 95% Solution B, 31.01 min with 5% Solution B, 40 min with 5% Solution B, and stop in 40.01 min) with a flow rate of 1 mL/min. rRNA was detected at 220 nm, and the area under the curve was integrated for the relative

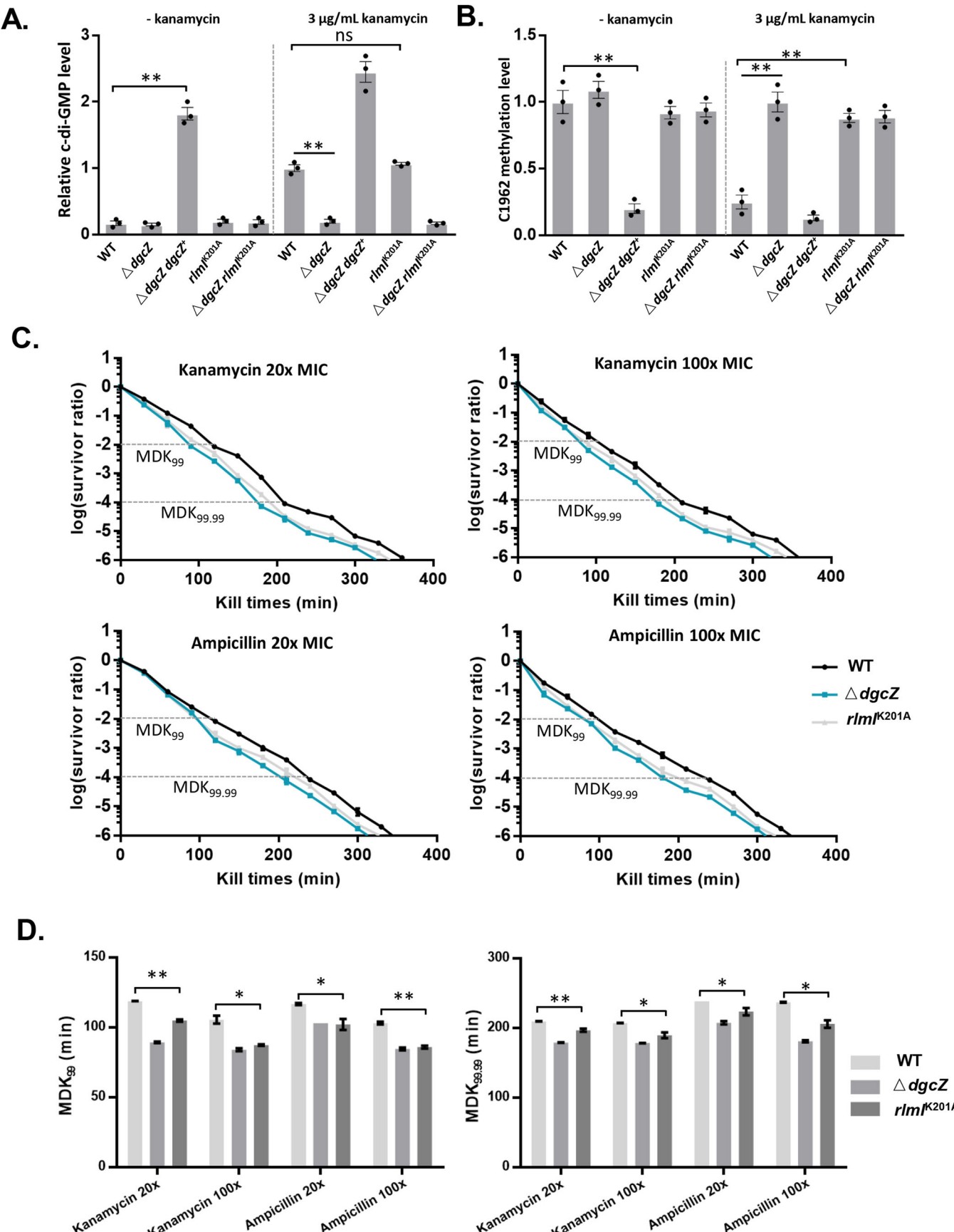

**Figure 6. c-di-GMP regulates ribosome assembly to promote antibiotic tolerance.**

(A) Relative c-di-GMP levels of the WT, Δ*dgcZ*, Δ*dgcZ dgcZ*$^+$, *rlmI*$^{K201A}$, and Δ*dgcZ rlmI*$^{K201A}$ strains. The intracellular c-di-GMP concentrations were determined by UPLC-IM-MS. The bar chart shows the relative quantification of c-di-GMP with the data points ($n = 3$ biological replicates, mean ± s.e.m.; **$p < 0.01$ ($p = 0.0014$ between WT and Δ*dgcZ dgcZ*$^+$; $p = 0.00083$ between WT and Δ*dgcZ*; $p = 0.27$ between WT and *rlmI*$^{K201A}$), two-tailed Student's t-test). (B) Methylation level of C1962 in 23S rRNA of the WT, Δ*dgcZ*, Δ*dgcZ dgcZ*$^+$, *rlmI*$^{K201A}$, and Δ*dgcZ rlmI*$^{K201A}$ strains. The endogenous methylation level in C1962 was determined by meRIP-qPCR in triplicate. The bar chart shows the relative quantification of methylated rRNA with the data points ($n = 3$ biological replicates, mean ± s.e.m.; **$p < 0.01$ ($p = 0.0043$ between WT and Δ*dgcZ dgcZ*$^+$; $p = 0.00095$ between WT and Δ*dgcZ*; $p = 0.00076$ between WT and *rlmI*$^{K201A}$), two-tailed Student's t-test). (C) Bacteria killing assays conducted on three bacterial strains using kanamycin and ampicillin. Experiments were performed at 20x and 100x the minimum inhibitory concentration (MIC) for each antibiotic to assess their efficacy in killing the bacteria ($n = 3$ biological replicates, mean ± s.e.m.). (D) Quantification of MDK$_{99}$ and MDK$_{99.99}$ values ($n = 3$ biological replicates, mean ± s.e.m.; **$p < 0.05$, **$p < 0.01$ (In the MDK$_{99}$ of WT and *rlmI*$^{K201A}$, $p = 0.0014$ in 20x MIC of kanamycin, $p = 0.025$ in 100x MIC of kanamycin, $p = 0.015$ in 20x MIC of ampicillin, $p = 0.0028$ in 100x MIC of ampicillin; In MDK$_{99.99}$ of WT and *rlmI*$^{K201A}$, $p = 0.0080$ in 20x MIC of kanamycin, $p = 0.014$ in 100x MIC of kanamycin, $p = 0.041$ in 20x MIC of ampicillin, $p = 0.014$ in 100x MIC of ampicillin). Source data are available online for this figure.

quantification of reaction products. This assay was performed in triplicate, and the results were calculated using GraphPad Prism 6.

## MeRIP-qPCR quantification of the endogenous C1962 methylation of 23S rRNA

Given the challenges with mass spectrometry, we established an alternative method using Methylated RNA Immunoprecipitation followed by quantitative PCR (MeRIP-qPCR). In this approach: 1. We synthesized RNA containing m5C1962 to serve as a standard for quantification. 2. To ensure the specificity of the antibodies used in MeRIP, we synthesized additional modified RNAs, including m2G1835, m2U2552, and m6A1618, as negative controls. This allowed us to evaluate potential cross-reactivity and confirm the antibody's ability to specifically recognize m5C modifications. 3. We tested three different m5C antibodies and determined that CST #28692 exhibited the highest specificity for m5C1962, with minimal cross-reactivity observed in the control experiments (Appendix Fig. S4).

*E. coli* strains (WT, Δ*dgcZ*, *rlmI*$^{K201A}$, Δ*dgcZ rlmI*$^{K201A}$) were cultivated overnight in LB medium at 37 °C and transferred to Vogel-Bonner medium at a 1:1000 ratio with 10 mM acetate and 3 μg/mL kanamycin at 25 °C for 24 h. 20 OD cells or 0.5 mg ribosomal subunit were harvested for RNA extraction using an RNA extraction kit (Sangon Biotech, China). Total RNA (50 μg) was diluted in 200 μL of IP buffer (20 mM HEPES, 50 mM KCl, and 1 U/μL protector RNase inhibitor, pH = 7.5) and sonicated (Sonics and Materials, USA) (20 cycles at 35% power, 15 s on/off) to prepare RNA fragments, which were divided into 50 μL for input and 150 μL for immunoprecipitation (IP). 2 μL anti-m5C antibody (#28692, CST, USA) was added to 150 μL of RNA fragments and incubated at 4 °C for 4 h. Then, 20 μL of protein A/G beads (Thermo Fisher Scientific, USA) were added to the mixture and incubated at 4 °C for 1 h. RNA was eluted with 0.1 M glycine (pH = 2) and neutralized with 1.0 M Tris (pH = 8). The eluted RNA and RNA fragments were purified with phenol-chloroform and quantified using a NanoDrop 2000 spectrophotometer (Thermo Fisher Scientific). Real-time RT-PCR was performed using a reverse transcription kit and real-time PCR kit (Applied Biosystems, USA) with the following primers.

| | Sequence1 (5′ → 3′) | Sequence2 (5′ → 3′) |
|---|---|---|
| C1962 | CGGTCCTAAGGTAGCGAAAT ACTGAGTCTCGGGTGGAGA | ACGGCGGCCGTAACTATA GCCTGGCCATCATTACGCC |

| | Sequence1 (5′ → 3′) | Sequence2 (5′ → 3′) |
|---|---|---|
| 23S rRNA | AGTGGAAGCGTCTGGAAAGG GCCCTACTCATCGAGCTCAC | ATCGTACCCCAAACCGACAC TTCTCCCGAAGTTACGGCAC |

## Measuring the relative mRNA levels of methyltransferases

*E. coli* strains (WT, *dgcZ*$^+$ and *dgcZ*$^{G206A,G207A}$) were cultivated overnight in LB medium at 37 °C, transferred to Vogel-Bonner medium at a 1:1000 ratio with 10 mM acetate at 25 °C for 12 h, and induced with 0.2 mM IPTG at 25 °C for 20 h. Cells corresponding to 10 OD were harvested for RNA extraction using an RNA extraction kit (Sangon Biotech, China). Real-time RT-PCR was performed using a reverse transcription kit and real-time PCR kit (Applied Biosystems, USA) with three replicates. The following primers were used.

| | Sequence1 (5′ → 3′) | Sequence2 (5′ → 3′) |
|---|---|---|
| *rlmI* | GAAGCGCTGGATATTGCACG ATCGCGATAAGTACGCAGCA | ACGTTTGACCCGTCTGAGTC GTCGAGGCCATCTTTTTGCG |
| *rlmE* | TGGCGCTAGAAATGTGTCGT ACCTTCGTAAACAGGGAGCG | GGCGCTAGAAATGTGTCGTG CCTTCGTAAACAGGGAGCGA |
| *gapA* | GATGGCCCGTCTCACAAAGA TGCCATTCAGTTCTGGCAGT | TTGACCTGACCGTTCGTCTG ACGTCATCTTCGGTGTAGCC |

## Analysis of ribosomal subunits by sucrose density gradient centrifugation

*E. coli* strains were cultivated overnight in LB medium at 37 °C and transferred to Vogel-Bonner medium at a 1:1000 ratio with 10 mM acetate and 3 μg/mL kanamycin at 25 °C for 24 h. Cells corresponding to 20 OD were harvested and subjected to three freeze-thaw cycles. The cells were treated with 1 mL of lysis buffer (20 mM HEPES, 0.5 mM or 10 mM MgCl$_2$, 200 mM NH$_4$Cl, 1 mg/mL lysozyme, 50 units/mL benzonase, pH 7.5) at 4 °C for 20 min with vigorous shaking. Following lysis, the mixture was centrifuged at $10,000 \times g$ for 10 min at 4 °C. The supernatant lysate was layered on top of a sucrose gradient (10–50%, *w/v*) in lysis buffer and separated by ultracentrifugation in a Beckman SW-28 Rotor at

**A.**

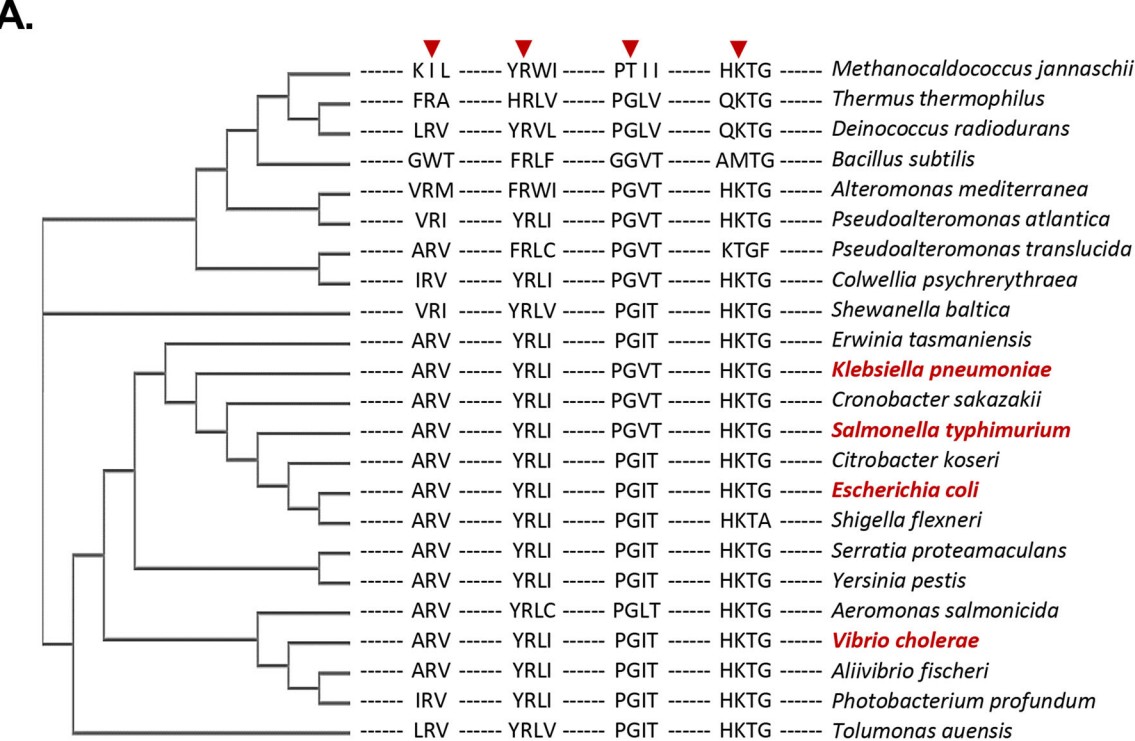

**B.**

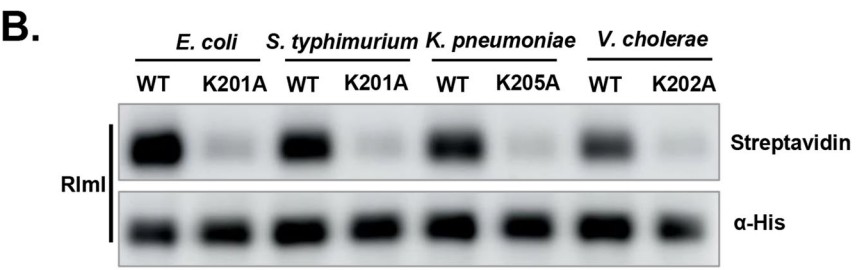

**C.**

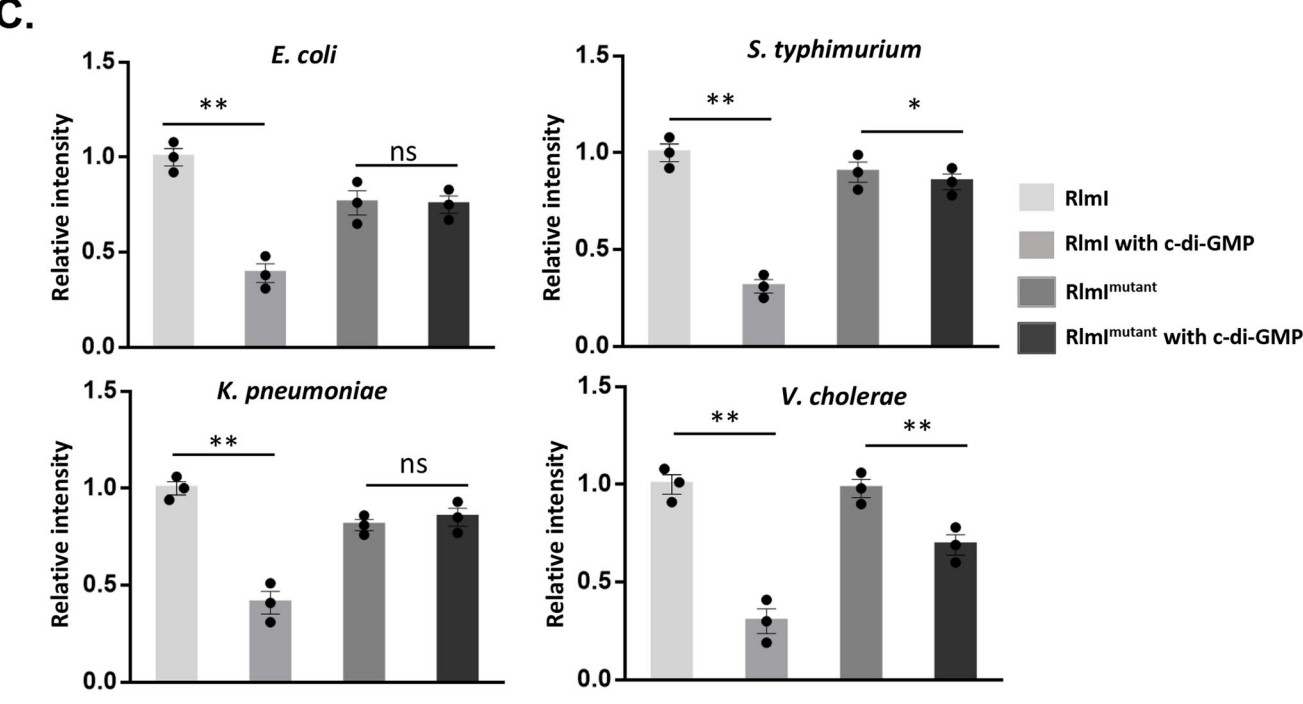

**Figure 7.   c-di-GMP binds RlmI and inhibits its activity conserved in other three pathogenic bacteria.**

(A) CLUSTALW alignment of the binding sites of c-di-GMP and RlmI. Residues involved in c-di-GMP binding (R64, R103, G114, and K201) are marked by red arrows. (B) Streptavidin blotting assays for RlmI of four species. The WT and mutant RlmI interacted with biotin-c-di-GMP and were crosslinked by UV. Streptavidin and α-His present interaction signals and protein levels, respectively. (C) In vitro methylation assay for the RlmI of four species. The experiment was performed in triplicate. The bar chart shows the quantitative results of methylation products with the data points ($n = 3$ biological replicates, mean ± s.e.m.; ns: no significant difference, *$p < 0.05$, **$p < 0.01$ (In *E. coli*, $p = 8.96E-5$ in RlmI, $p = 0.62$ in mutated RlmI; In *S. typhimurium*, $p = 0.00028$ in RlmI, $p = 0.049$ in mutated RlmI; In *K. pneumoniae*, $p = 0.0015$ in RlmI, $p = 0.15$ in mutated RlmI; In *V. cholerae*, $p = 0.00048$ in RlmI, $p = 0.00040$ in mutated RlmI), two-tailed Student's t-test). Source data are available online for this figure.

120,000 × $g$ for 12 h at 4 °C. The suspension was recovered to 25 components and quantified by measuring absorbance at 260 nm using a NanoDrop 2000 spectrophotometer.

## Molecular dynamics simulations

Microsecond-scale molecular dynamics (MD) simulations were employed in this study (Frenkel and Smit, 2001). In MD simulations, the corresponding Newton's equations of motion are numerically integrated after selecting appropriate force fields, enabling the monitoring of each individual atom in various systems, including liquids at interfaces, solid walls, and biological membranes (Marrink et al, 2019; Nagy et al, 2007). The number of particles, pressure, and temperature of the system were fixed, while the volume was adjusted accordingly. MD simulations can model hydrogens at the classical or quantum levels (Calero et al, 2015). Beyond energetic and structural properties, MD simulations provide access to time-dependent quantities such as diffusion coefficients or spectral densities, enhancing its applicability. In this study, we conducted MD simulations of c-di-GMP bound to rRNA methyltransferases (RlmI). The simulation system included one rRNA methyltransferase and seven c-di-GMP molecules, fully solvated by 21,818 TIP3P water molecules in a potassium chloride solution at a concentration of 0.15 M, yielding a system size of 68,532 atoms. In each of the two statistically independent MD simulations, a single c-di-GMP molecule was positioned at the center of the simulation box, near domains I, II, and III of RlmI, while the remaining six free c-di-GMPs were randomly distributed around the RlmI protein according to the default settings of the Charmm-GUI "Multicomponent Assembler". All MD simulation inputs were generated using the CHARMM-GUI platform (Jo et al, 2008; Kern et al, 2022), and the CHARMM36m force field (Huang and MacKerell, 2013) was adopted for interactions between rRNA methyltransferases and c-di-GMP. The force field used also included the parameterization of the species c-di-GMP. All hydrogen bonds were maintained at fixed lengths, allowing fluctuations bond distances and angles for other atoms. The crystal structure of the rRNA methyltransferases was downloaded from the RCSB PDB Protein Data Bank (file name "3c0k"). The system was energy-minimized for 50,000 steps and well-equilibrated (NVT equilibration in Figure 16 of the SI) for 250 ps before the MD simulation was performed. Production runs were performed with an NPT ensemble for 2 μs. The pressure and temperature were set at 1 atm and 310.15 K, respectively, to simulate the human body environment. The GROMACS 2021 package was utilized for all MD simulations (Berendsen et al, 1995). Time steps of 2 fs were used in the production simulations, and the particle mesh Ewald method with a Coulomb radius of 1.2 nm was employed to compute long-range electrostatic interactions. The cut-off for

Lennard-Jones interactions was set to 1.2 nm. Pressure was controlled with a Parrinello-Rahman piston with a damping coefficient of 5 ps$^{-1}$, while temperature was controlled with a Nosé-Hoover thermostat with a damping coefficient of 1 ps$^{-1}$. Periodic boundary conditions in three directions of space were considered. We employed the "gmx-sham" tool of the GROMACS 2021 package to perform Gibbs free energy landscape analysis. Moreover, the software VMD (Humphrey et al, 1996) and UCSF Chimera (Pettersen et al, 2004) were used for trajectory analysis and visualization.

The radius of gyration $R_g$, used as a reaction coordinate in the computation of Gibbs free energy landscapes, was determined as follows:

$$Rg = \sqrt{\frac{\sum_i ||r_i||^2 m_i}{\sum_i m_i}} \tag{1}$$

where $m_i$ is the mass of atom $i$, and $r_i$ is the position of the same atom with respect to the center of mass of the selected group. The RMSD was calculated as follows:

$$RMSD(t) \equiv \sqrt{\frac{1}{N}\sum_{i=1}^{N}\delta_i^2(t)} \tag{2}$$

where $\delta_i$ is the difference in distance between atom $i$ [located at $x_i(t)$] of the catalytic domain and the equivalent location in the crystal structure. The RMSF values were obtained as follows:

$$RMSFi \equiv \sqrt{\sum_{t_j=1}^{\Delta t}(x_i(t_j) - \widetilde{x}_i)^2},$$

where $\widetilde{x}_i$ is the time average of $x_i$ and where $\Delta t$ is the time interval at which the average was taken.

## ITC assay

In the isothermal titration calorimetry (ITC) assay, c-di-GMP (#C057 of Biolog), RlmI, and its mutants were prepared in a titration buffer (20 mM Tris, 50 mM NaCl, 200 mM KCl, pH 7.0). Protein concentrations were determined based on Coomassie brilliant blue staining. ITC titrations were performed using a MicroCal iTC200 system (GE Healthcare, PA, USA) at 25 °C. Each titration consisted of 22 injections of 5 μL c-di-GMP. A stock solution of 0.5 mM c-di-GMP was titrated into WT or mutant RlmI (25 μM) in sample cells of 200 μL volume individually. c-di-GMP was titrated into 200 μL of the titration buffer as a control for data processing. The resulting titration curves were processed using the Origin 7.0 software program (OriginLab) according to the "one set of sites" fitting model.

## Streptavidin blotting assay

In this assay, RlmI (0.1 mg/mL) and its mutants were incubated with 10 μM biotin-c-di-GMP in a reaction buffer (20 mM Tris, 50 mM NaCl, 200 mM KCl, pH 7.0) at 37 °C for 1 h. The samples were subjected to UVcross-linking on ice for 0.5 h to further link c-di-GMP to RlmI. These linked samples were divided into two parts for western blot analysis. After incubation with IRDye 800CW Conjugated Streptavidin (#926-32230; LI-COR Biosciences, USA) at room temperature for 2 h, another membrane was incubated with an anti-His antibody (05-949, Millipore, USA) at 4 °C for 12 h and then incubated with an IRDye 800 secondary antibody for 1 h. The resulting membranes were visualized with an Odyssey Infrared Imaging System (LI-COR Biosciences).

## Isolation and quantification of c-di-GMP in *E. coli*

Isolation of c-di-GMP was conducted as previously described (Spangler et al, 2010; Xu et al, 2019). Briefly, *E. coli* cells at 50 OD were harvested and resuspended in 2 mL of ddH$_2$O. Subsequently, 8 mL of a 50% methanol and 50% acetonitrile mixture was added to extract intracellular c-di-GMP. Moreover, 1 μM cGMP was added as an internal reference. For absolute quantification of c-di-GMP, the density of *E. coli* sediment was defined as 1 mg/mL, and bacterial concentration was calculated using absorbance measurements, which were used for c-di-GMP quantification. The extracts were analyzed via ultrahigh-performance liquid chromatography coupled with ion mobility mass spectrometry (UPLC-IM-MS), utilizing a Waters UPLC I-class system equipped with a binary solvent delivery manager and a sample manager coupled with a Waters VION IMS Q-TOF mass spectrometer equipped with an electrospray interface (Waters Corporation, CT, USA).

## Determination of the strain growth curve in Vogel-Bonner medium

As previously mentioned, the strains WT, Δ*dgcZ*, *rlmI*[K201A], and Δ*dgcZ rlmI*[K201A] were grown in Vogel-Bonner medium supplemented with 10 mM acetate at 25 °C. For kanamycin treatment, concentrations of 0, 1.5, 3, 6, or 9 μg/mL kanamycin were added to the Vogel-Bonner medium Cell concentrations were measured at OD$_{600}$ using a NanoDrop 2000 spectrophotometer at 8, 12, 16, 24, and 32 h. The growth curve was subsequently plotted using GraphPad Prism 6.

## Kanamycin and ampicillin killing assay

To evaluate the killing kinetics of kanamycin and ampicillin, bacterial cultures were grown to mid-log phase (OD600: 0.4) in LB medium. Equal volumes of bacterial suspensions were distributed into 96-well plates in LB medium. Each well received varying concentrations containing kanamycin (MIC: 8 μg/mL) or ampicillin (MIC: 5 μg/mL), specifically at 20× and 100× MIC. The plates were incubated at 37 °C at 200 rpm for a total duration of 6 h, with measurements taken every 30 min. After each time point, aliquots were collected from each well, serially diluted in PBS, and plated on LB agar for overnight incubation to determine colony-forming units (CFUs). Each measurement was conducted in triplicate, and the average CFU count was calculated. The reduction in CFUs

compared with the control was used to evaluate the killing efficiency of each antibiotic. All experiments were independently repeated three times, and the killing curve was plotted using GraphPad Prism 6.

## Statistical analysis

Pairwise comparisons were performed using two-tailed Student's *t*-test, and statistical significance was set at $*p < 0.05$ and $**p < 0.01$. Error bars represent the means ± s.e.m.s.

# Data availability

The crystal structure files, MD simulation files (input files, parameter files, topology files, etc.), and structures of c-di-GMP are available on the website https://github.com/Zheyao-Hu/RlmIcdiGMP. Moreover, all the software (free to use) packages used in this study were the official release versions without any modifications. The raw protein microarray data have been published in the Protein Microarray Database (www.proteinmicroarray.cn/) with the accession number PMDE226.

The source data of this paper are collected in the following database record: biostudies:S-SCDT-10_1038-S44319-025-00377-w.

# Peer review information

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

## Acknowledgements

This study was supported by the National Natural Science Foundation of China (Grant No. 32000027), the Natural Science Foundation of Fujian Province, China (No. 2022J01197), the Fourteenth Five-Year National Key Research and Development Program of China (2023YFC2307200), the R&D Program of Guangzhou National Laboratory (No. GZNL2023A01005) and the National Natural Science Foundation of China (No. 92374110). The Public Technology Service Center, Fujian Medical University provides instrument support.

## Author contributions

**Siqi Yu**: Conceptualization; Data curation; Formal analysis; Validation; Investigation; Methodology; Writing—original draft; Writing—review and editing. **Zheyao Hu**: Conceptualization; Data curation; Formal analysis; Investigation; Visualization; Methodology; Writing—original draft; Writing—review and editing. **Xiaoting Xu**: Formal analysis; Validation; Investigation; Writing—review and editing. **Xiaoran Liang**: Investigation; Visualization. **Jiayi Shen**: Investigation; Visualization. **Min Liu**: Investigation; Visualization. **Mingxi Lin**: Investigation; Visualization. **Hong Chen**: Investigation; Visualization; Methodology. **Jordi Marti**: Conceptualization; Supervision; Methodology; Writing—review and editing. **Sheng-ce Tao**: Conceptualization; Funding acquisition; Writing—review and editing. **Zhaowei Xu**: Conceptualization; Supervision; Funding acquisition; Investigation; Writing—original draft; Project administration; Writing—review and editing.

Source data underlying figure panels in this paper may have individual authorship assigned. Where available, figure panel/source data authorship is listed in the following database record: biostudies:S-SCDT-10_1038-S44319-025-00377-w.

## Disclosure and competing interests statement

The authors declare no competing interests.

