## [Peer Review File · EMBO Reports]

c-di-GMP inhibits rRNA methylation and impairs ribosome assembly in the presence of kanamycin

Siqi Yu, Zheyao Hu, Xiaoting Xu, Xiaoran Liang, Jiayi Shen, Min Liu, Mingxi Lin, Hong Chen, Jordi Marti, Sheng-ce Tao, and Zhaowei Xu

Corresponding author(s): Zhaowei Xu (xuzw@fjmu.edu.cn) , Sheng-ce Tao (taosc@sjtu.edu.cn), Jordi Marti (jordi.marti@upc.edu)

Review Timeline:

Transfer Date:	18th Jul 24
Editorial Decision:	24th Jul 24
Revision Received:	30th Sep 24
Editorial Decision:	5th Nov 24
Revision Received:	21st Nov 24
Editorial Decision:	19th Dec 24
Revision Received:	20th Dec 24
Accepted:	15th Jan 25

Editor: Esther Schnapp

Transaction Report: This manuscript was transferred to EMBO reports following peer review at The EMBO Journal.

Referee #1:

The manuscript by Siqi Yu and colleagues presents interesting results on the interaction of the second messenger c-di-GMP with 23S rRNA methylases RlmI and RlmE. Previously, c-di-GMP was implicated primarily in regulating extracellular and cell envelope-related processes, such as protein and polysaccharide secretion, biofilm formation, cell division, and motility. This paper extends the list of c-di-GMP-controlled functions to ribosome assembly. This work is an important development and, if true, would open a new avenue in cell research and could even help in the search for new antibiotics. Having said that, this work misses some essential controls that are necessary for proving that the observed effects are physiologically relevant and specific for c-di-GMP. On the other hand, the extensive discussion of the structure of the putative RlmI-c-di-GMP complex and molecular dynamics simulations of its behavior are presented as if the existence of such a complex has already been proven. This is not the case, which makes the paper look incomplete and overextended.

General comments

1. The 23S rRNA methylases RlmI and RlmE were identified as c-di-GMP binders based on a screen of *E. coli* proteins for their interaction with biotin-linked c-di-GMP. The *E. coli* cell is full of various nucleotides, including ATP, cAMP, GTP, and ppGpp, which are typically present at much higher (up to millimolar) levels than c-di-GMP and could potentially interfere with c-di-GMP binding. Proving that a certain protein specifically binds c-di-GMP requires testing that its c-di-GMP binding is not prevented by a 100-fold excess of such nucleotides. Unfortunately, this has not been done, all experiments have been conducted in vitro with c-di-GMP as the sole effector, which provokes the question of whether the observed effects are caused by c-di-GMP (and not GTP, GMP, or cGMP) and are specific for it.

2. A substantial part of the paper (lines 432-500, Fig. 3A-C, Fig. 4A-C, Fig. S5 through S15, supplementary movies 1 and 2) is devoted to the prediction of the structure of the putative RlmI-c-di-GMP complex and molecular dynamics simulations of its properties. I believe that in the absence of a high-resolution structure of this proposed complex, prediction of the c-di-GMP-binding residues of RlmI (except for experimentally tested R64, R103, G114, and K201) and molecular dynamics simulations of its behavior are premature and highly speculative.

3. Lines 91 - 100 discuss the antibiotic resistance caused by methylation of 16S rRNA and A2503 in 23S rRNA. It is not clear how this is relevant for this work, which studied RlmI and RlmE, which are not known (at least to me) to be linked with antibiotic resistance. Antibiotic resistance does not seem to be related to the issues addressed in this work.

4. The paper repeatedly mentions four methyltransferases (lines 103, 353, 394, 395, 592) but shows the results only for two, RlmI and RlmE. What were the other two methyltransferases and were the c-di-GMP binding results similar for them as well?

Specific comments

- Fig. 1B,C. The graphs are presented without proper explanation. Is it the second peak that is being

measured in Fig. 1D? Which compound does it represent?

- Although Fig. 3 is entitled "Binding model", its legend does not explain how was this model derived. Instead, the proposed binding model is presented as if it were real. With its RMSD and RMSF values and predicted loop closure (domain III) upon binding, the figure gives the false impression of reflecting several crystal structures. The binding mode might indeed be as depicted in the figure but it might be entirely different; the proposed loop could be happening in vivo, or it might not, and there is no proof that either of the panels reflects actual events.
- Same with Fig. 4, panels A-C. Although this figure is annotated as "Determination of the binding sites of c-di-GMP on RlmI", these panels represent possible, albeit unproven, results of a molecular dynamics simulation of the putative RmlI-c-di-GMP complex, whose structure - and very existence - has not been proven in the first place. These three panels are all highly speculative and are not based on experimental results, which are presented in Fig. 4D, E, and F.
- Fig. 4E shows the ITC titration data of RmlI with K_d for c-di-GMP of 1.3 μM . However, the description of the ITC process (lines 294-297) mentions the injections of 5 μL of c-di-GMP from the stock solution of 0.5 mM c-di-GMP into a 200 μL chamber. That would make the c-di-GMP concentration of 12.5 μM after the very first injection and it would increase by 12.5 μM with every subsequent injection. How could this result in K_d of 1.3 μM ?
- Fig. 6A presents a list of 23 organisms, connected by some sort of a dendrogram. What does this dendrogram represent, a 16S rRNA-based tree, an RlmI tree, or something else? Given that c-di-GMP binding likely involves other residues besides R64, R103, G114, and K201 (as depicted in Fig. 4C), it is probably not enough to check just for these four residues to conclude that their conservation reflects the ability of the RlmI's from multiple pathogenic bacteria to bind c-di-GMP.
- Figures S3 and S4 are unnecessary. Most readers of this journal probably already know the structures of c-di-GMP and ten common amino acids.

In conclusion, several key issues need to be addressed before this paper can be accepted. The ability of RlmI to bind c-di-GMP should be tested for its specificity with respect to other (di)nucleotides. The 3D structure of the predicted RmlI-c-di-GMP complex must be solved and the binding residues must be identified from the structure, rather than predicted by some unknown method. Only after that the molecular dynamics simulations would make sense.

Referee #2:

In this study, the authors demonstrate that two 23S rRNA methyltransferases, RlmI and RlmE in *E. coli* bind to the ubiquitous bacterial second messenger cyclic-di-GMP. They show that c-di-GMP binding inhibits RlmI methyltransferase activity, influencing ribosome assembly and consequently promoting antibiotic resistance. Molecular dynamic simulations, supported by site specific mutagenesis, were used to characterise the c-di-GMP binding site and identify residues essential for dinucleotide binding. The authors then showed that c-di-GMP binding to RlmI is conserved in antimicrobial resistant *E. coli* strains as well as in several different pathogenic species. This is an interesting and well conducted study that presents a novel and exciting function for c-di-GMP. I have a few relatively minor comments on the manuscript as it stands:

- Line 69: "RsmA fulfils quality control requirements in the last stages of small subunit assembly [8]" To avoid confusion, it might be worth noting that RsmA is referred to elsewhere in the literature (including reference 8) as KsgA.
- Line 391-393: This should probably say 'the kanamycin induced increase in c-di-GMP levels', as results are not shown for exogenous c-di-GMP production absent antibiotics.
- Line 401: This conclusion should be modified to reflect the fact that RlmI inactivation appears to be behind c-di-GMP induced ribosome assembly delay. As it stands, it reads as if c-di-GMP induces ribosome assembly, as opposed to inhibiting it.
- Line 510: This sentence is difficult to parse and should be rephrased for clarity.
- Line 519: 'As a close correlation exists between c-di-GMP...'
- Line 546: This sentence could be truncated, with '...mutated the c-di-GMP binding site K201 of RlmI (rlmIK201A)...' removed. The K201A mutants have already been introduced earlier in this section and their construction doesn't need to be stated again.
- Figure 5C: The Y Axis for these growth curves presumably shows OD (600nm), not absorbance. This should be changed for all four graphs.
- Line 558: The authors name the 13/40 strains that are Kan resistant, but these do not include C12, which is mentioned on Line 562 as being one that could not be mutated. This discrepancy should be clarified.
- Some of the references are misassigned. For example, while the journals and titles appear to be correct, references 48 and 49 are attributed to Hayes, C. S. et al. and Søggaard-Andersen, L. et al. These references should be attributed to Little et al. (PLOS Genetics 2016) and Grenga et al. (PLOS Genetics 2020) respectively. These were the only two errors I found, but the authors should still carefully check the reference list and ensure that all details are correct in each case.
- Line 741: The rlmI mutant construction details should be moved to the M&M section.
- Line 788 onwards: Again, some of the descriptions in this figure legend (e.g. growth conditions for strains) should be moved to the M&M section.

Referee #3:

Yu et al follow up on their recent proteomic study which implicated c-di-gmp as binding partner with methyltransferases that are involved in ribosome assembly. They provide genetic and biochemical evidence for an impact of c-di-gmp in inhibiting ribosome assembly and try to establish role in antibiotic tolerance or antibiotic resistance. Authors show that c-di-gmp mediated inhibition of the methyltransferase RmlI is conserved in multiple pathogenic bacteria.

Major concerns:

1. It is unclear what the physiological role of c-di-gmp mediated inhibition of ribosome assembly is. A role in antibiotic tolerance is explored but it does not explain the widespread impact of c-di-gmp mediated inhibition of ribosome assembly. A potential general impact on growth does not occur which complicates the establishment of a general role in a stress response. In the end, authors only observe weak growth effects during kanamycin treatment.

2. The concentrations of c-di-gmp that are required to inhibit the methyltransferase in an in vitro reconstituted system seems to be very high (micro molar range). While overexpression effects and loss of function effects of c-di-gmp are shown it seems that there is potential for the involvement of additional factors in the observed phenomenon or that it is an in vitro artifact. In the absence of a demonstrated physiological role this leads to uncertainty with the main findings of this study.

3. Authors use confusing terminology when they describe the impact of antibiotic treatment on the increased expression of c-di-gmp and corresponding inhibition of ribosome assembly, leading to an inaccurate and misleading description of antibiotic tolerance and antibiotic resistance and to unclear distinctions between alternative methyltransferases that mediate aminoglycoside resistance in resistant isolates (post ribosome assembly) and methyltransferases that govern ribosome assembly (pre-assembly). The focus of the paper starts out on pre-assembly thus the sudden connection that the authors try to establish to antibiotic resistance is confusing. A general role on antibiotic tolerance via growth inhibition seems more likely but the overexpression strain does not inhibit growth. To be clear: If the target of the antibiotic (assembled ribosome) is modified it will promote antibiotic resistance, leading to a clearly elevated minimal inhibitory concentration of the antibiotic in terms of growth inhibition, whereas if the ribosome assembly itself is inhibited it may promote antibiotic tolerance because of the inactivity of the target. Authors do not clearly distinguish between both options and more importantly they do not provide evidence for one or the other option. Antibiotic resistance in terms of minimal inhibitory concentrations with kanamycin is not shown to occur. The growth curves only show a reduced growth rate in the c-di-gmp mutants, they do not show growth inhibition. If the authors want to establish a role in antibiotic resistance they have to demonstrate an impact on the minimal inhibitory concentration. If they want to demonstrate a role on antibiotic tolerance they have to present biphasic killing kinetics that support that role. Given the pleiotropic role of c-di-gmp the impact on growth rate during antibiotic treatment cannot be tied directly to its impact on the interaction with the methyltransferase. Without an established physiological role for this phenomenon of c-di-gmp inhibition of pre-assembly methyltransferases and with these unclear findings authors are not able to establish a mechanism that would explain their observations. The authors would be better off to exclude this part and focus on the physiological mechanism and role of c-di-gmp mediated inhibition of ribosome assembly.

Minor concerns:

1. Language throughout the manuscript should be revised. There are a number of grammar mistakes and the manuscript is at times not concise enough, especially towards the end of the manuscript when the impact of antibiotic treatment is explored

2. Isolation method of E. coli strains. Seems that this is solely based on growth on MacConkey Agar which is way too vague. hardly any information on the clinical isolates is provided. A better characterization of the isolates is necessary and should include MICs via Vitek etc. But given the unclear findings it is questionable it does not really make a difference

3. how were the strains normalized to quantify ci-di-gmp content - did the authors determine CFUs/ml?

l 381: was controlled is misleading. authors could say has been shown to be controlled by...

l 385: effector? seems right for c-di-gmp not DgcZ

l389: terminology: replenishment strain?

l510: grammar

l540: grammar

l545: grammar and overstatement

l549: grammar

l552: grammar

l567: overstatement

l593: overstatement

l597: overstatement

l599: overstatement

l614: what is an upstream regulatory signal and how was this established?

l631: conflating pre-methyltransferases with post-methyltransferases

Dear Dr. Xu

Thank you for the transfer of your manuscript with referee reports to EMBO reports. We think that it is potentially a good fit for our journal.

As we discussed, we would like to ask you to please address referee 1's point 1, make clear that the model is just one possible structure (probably out of many), and address point 3 by referee 3 to the best of your abilities. In general, it would be good if you could explain better, or provide more data on, the physiological role of c-di-GMP-mediated inhibition of ribosome assembly. All other minor points will also need to be addressed.

I would thus like to invite you to revise your manuscript with the understanding that the referee concerns must be fully addressed and their suggestions taken on board, as indicated above. Please address all referee concerns in a complete point-by-point response. Acceptance of the manuscript will depend on a positive outcome of a second round of review. It is EMBO reports policy to allow a single round of major revision only and acceptance or rejection of the manuscript will therefore depend on the completeness of your responses included in the next, final version of the manuscript.

We realize that it is difficult to revise to a specific deadline. In the interest of protecting the conceptual advance provided by the work, we recommend a revision within 3 months (24th Oct 2024). Please discuss the revision progress ahead of this time with us if you require more time to complete the revisions.

- 1) A data availability section providing access to data deposited in public databases is missing. If you have not deposited any data, please add a sentence to the data availability section that explains that.
- 2) Your manuscript contains statistics and error bars based on $n=2$. Please use scatter blots in these cases. No statistics should be calculated if $n=2$.

3) We replaced Supplementary Information with Expanded View (EV) Figures and Tables that are collapsible/expandable online. A maximum of 5 EV Figures can be typeset. EV Figures should be cited as 'Figure EV1, Figure EV2' etc... in the text and their respective legends should be included in the main text after the legends of regular figures.

5) a complete author checklist, which you can download from our author guidelines <https://www.embopress.org/page/journal/14693178/authorguide>. Please insert information in the checklist that is also reflected in the manuscript. The completed author checklist will also be part of the RPF.

6) Please note that all corresponding authors are required to supply an ORCID ID for their name upon submission of a revised manuscript (<https://orcid.org/>). Please find instructions on how to link your ORCID ID to your account in our manuscript tracking system in our Author guidelines <https://www.embopress.org/page/journal/14693178/authorguide#authorshipguidelines>

12) All Materials and Methods need to be described in the main text using our 'Structured Methods' format, which is required for all research articles. According to this format, the Methods section includes a Reagents and Tools Table (listing key reagents, experimental models, software and relevant equipment and including their sources and relevant identifiers) followed by a Methods and Protocols section describing the methods using a step-by-step protocol format. The aim is to facilitate adoption of the methodologies across labs. More information on how to adhere to this format as well as a downloadable template (.docx) for the Reagents and Tools Table can be found in our author guidelines:
<https://www.embopress.org/page/journal/14693178/authorguide#structuredmethods>.

An example of a Method paper with Structured Methods can be found here: <https://www.embopress.org/doi/full/10.1038/s44320-024-00037-6#sec-4>

I look forward to seeing a revised form of your manuscript when it is ready.

Referee #1:

General comments

1. The 23S rRNA methylases RlmI and RlmE were identified as c-di-GMP binders based on a screen of *E. coli* proteins for their interaction with biotin-linked c-di-GMP. The *E. coli* cell is full of various nucleotides, including ATP, cAMP, GTP, and ppGpp, which are typically present at much higher (up to millimolar) levels than c-di-GMP and could potentially interfere with c-di-GMP binding. Proving that a certain protein specifically binds c-di-GMP requires testing that its c-di-GMP binding is not prevented by a 100-fold excess of such nucleotides. Unfortunately, this has not been done, all experiments have been conducted in vitro with c-di-GMP as the sole effector, which provokes the question of whether the observed effects are caused by c-di-GMP (and not GTP, GMP, or cGMP) and are specific for it.

Response: We thank the reviewer for this insightful comment. To verify the specificity of the interaction between RlmI and c-di-GMP, we conducted cross-linking and immunoblotting experiments to detect this interaction. In these assays, we used the 100-fold excess of unlabelled c-di-GMP and its analogues (GTP, GMP, ATP, AMP, cAMP and ppGpp) as competitive inhibitors to observe their effects on the interaction between RlmI and biotinylated c-di-GMP.

The results demonstrated that unlabelled c-di-GMP effectively blocked the interaction between biotinylated c-di-GMP and RlmI, whereas none of the other analogues exhibited a similar effect. This finding confirms that the interaction between RlmI and c-di-GMP in *Escherichia coli* is indeed specific.

We have updated the relevant sections to reflect these results. (lines 398-405)

Specific comments

- Fig. 1B,C. The graphs are presented without proper explanation. Is it the second peak that is being measured in Fig. 1D? Which compound does it represent?

Response: We apologize for the confusion. Yes, the second peak represents the methylated rRNA, which is used to calculate the activity of the methyltransferase. To enhance clarity, we have marked this peak with the black arrow and included an explanation in the figure legend.

- Although Fig. 3 is entitled "Binding model", its legend does not explain how was this model derived. Instead, the proposed binding model is presented as if it were real. With its RMSD and RMSF values and predicted loop closure (domain III) upon binding, the figure gives the false impression of reflecting several crystal structures. The binding mode might indeed be as depicted in the figure but it might be entirely different; the proposed loop could be happening in vivo, or it might not, and there is no proof that either of the panels reflects actual events.

Response: Thank you for pointing this out. We have clarified that the binding model is based on molecular dynamics simulations rather than co-crystallization data. We have revised the figure and added details about the model's source in the figure legend.

- Same with Fig. 4, panels A-C. Although this figure is annotated as "Determination of the binding sites of c-di-GMP on RImI", these panels represent possible, albeit unproven, results of a molecular dynamics simulation of the putative RImI-c-di-GMP complex, whose structure - and very existence - has not been proven in the first place. These three panels are all highly speculative and are not based on experimental results, which are presented in Fig. 4D, E, and F.

Response: Thank you for your comment. In this section, our aim was to identify the interaction sites of c-di-GMP with RImI. We first employed molecular dynamics simulations to locate these interaction sites, followed by experimental validation of the identified sites. Through the simulations, we identified six potential interaction sites, and experimental data confirmed that four of these are key interaction sites between c-di-GMP and RImI.

To clarify, we have added explanations in both the main text and the figure legend for Figures 4A-C, indicating that these results are based on molecular dynamics simulations.

- Fig. 4E shows the ITC titration data of RImI with K_d for c-di-GMP of 1.3 μM . However, the description of the ITC process (lines 294-297) mentions the injections of 5 μL of c-di-GMP from the stock solution of 0.5 mM c-di-GMP into a 200 μL chamber. That would make the c-di-GMP concentration of 12.5 μM after the very first injection and it would increase by 12.5 μM with every subsequent injection. How could this result in K_d of 1.3 μM ?

Response: Thank you for this observation. In ITC experiments, the concentration of c-di-GMP in the syringe should be approximately 20 times that of the protein concentration (25 μM RImI in this assay) to ensure binding saturation before half of the titration (PMID: 28889310). The interaction affinity is calculated based on the fitted heat release curve.

- Fig. 6A presents a list of 23 organisms, connected by some sort of a dendrogram. What does this dendrogram represent, a 16S rRNA-based tree, a RImI tree, or something else? Given that c-di-GMP binding likely involves other residues besides R64, R103, G114, and K201 (as depicted in Fig. 4C), it is probably not enough to check just for these four residues to conclude that their conservation reflects the ability of the RImI's from multiple pathogenic bacteria to bind c-di-GMP.

Response: Thank you for your comments. The figure presents a phylogenetic tree based on the RImI protein sequences. Based on the results of this analysis, we selected RImI proteins from three species for interaction validation. These results indicate that c-di-GMP can interact with the RImI proteins from *S. typhimurium*, *K. pneumoniae*, and *V. cholerae* strains. To clarify, we have revised the results to "the effect of c-di-GMP on RImI may be conserved in multiple pathogenic bacteria". (Lines 581-582)

- Figures S3 and S4 are unnecessary. Most readers of this journal probably already know the structures of c-di-GMP and ten common amino acids.

Response: We appreciate the reviewer's suggestion. Figures S3 and S4 have been removed as requested.

Referee #2:

• **Line 69: "RsmA fulfils quality control requirements in the last stages of small subunit assembly [8]" To avoid confusion, it might be worth noting that RsmA is referred to elsewhere in the literature (including reference 8) as KsgA.**

Response: We thank the reviewer for this comment. It has been modified to read "RsmA, also known as KsgA, fulfills quality control requirements in the last stages of small subunit assembly".

• **Line 391-393: This should probably say 'the kanamycin induced increase in c-di-GMP levels', as results are not shown for exogenous c-di-GMP production absent antibiotics.**

Response: We appreciate the reviewer's suggestion and have amended it to "the increase in c-di-GMP levels induced by kanamycin inhibited the assembly of large ribosomal subunits in *E. coli*". (lines 387-389)

• **Line 401: This conclusion should be modified to reflect the fact that RlmI inactivation appears to be behind c-di-GMP induced ribosome assembly delay. As it stands, it reads as if c-di-GMP induces ribosome assembly, as opposed to inhibiting it.**

Response: We thank the reviewer for pointing this out. As suggested, the description has been revised to "c-di-GMP inhibits ribosome assembly by inactivating RlmI".

• **Line 510: This sentence is difficult to parse and should be rephrased for clarity. We found that the stoichiometries have not significant different between RlmI and its variants, which showed 1:1 binding ratio with c-di-GMP (Fig. 4E).**

Response: For clarity, it has been rephrased to, "we found that the stoichiometry of RlmI and its variants with c-di-GMP is not significantly different, each showing a 1:1 binding ratio".

• **Line 519: 'As a close correlation exists between c-di-GMP...'**

Response: We thank the reviewer for this comment. The sentence has been revised as suggested.

• **Line 546: This sentence could be truncated, with '...mutated the c-di-GMP binding site K201 of RlmI (rlmIK201A)...' removed. The K201A mutants have already been introduced earlier in this section and their construction doesn't need to be stated again.**

Response: The repeated mention of the K201A mutant construction has been removed.

- **Figure 5C: The Y Axis for these growth curves presumably shows OD (600nm), not absorbance. This should be changed for all four graphs.**

Response: We thank the reviewer for pointing this out. The Y-axis of the four graphs in Figure 5C has been modified to OD (600 nm).

- **Line 558: The authors name the 13/40 strains that are Kan resistant, but these do not include C12, which is mentioned on Line 562 as being one that could not be mutated. This discrepancy should be clarified.**

Response: We have removed the MIC-related results and added the bacteria killing curves.

- **Some of the references are misassigned. For example, while the journals and titles appear to be correct, references 48 and 49 are attributed to Hayes, C. S. et al. and Sjøgaard-Andersen, L. et al. These references should be attributed to Little et al. (PLOS Genetics 2016) and Grenga et al. (PLOS Genetics 2020) respectively. These were the only two errors I found, but the authors should still carefully check the reference list and ensure that all details are correct in each case.**

Response: Thank you for your careful review. We have corrected references 48 and 49 and ensured the accuracy of the other references throughout the manuscript.

- **Line 741: The rlmI mutant construction details should be moved to the M&M section.**

Response: We have made the suggested revisions.

- **Line 788 onwards: Again, some of the descriptions in this figure legend (e.g. growth conditions for strains) should be moved to the M&M section.**

Response: We have revised the manuscript as suggested.

Referee #3:

Major concerns:

3. Authors use confusing terminology when they describe the impact of antibiotic treatment on the increased expression of c-di-gmp and corresponding inhibition of ribosome assembly, leading to an inaccurate and misleading description of antibiotic tolerance and antibiotic resistance and to unclear distinctions between alternative methyltransferases that mediate aminoglycoside resistance in resistant isolates (post ribosome assembly) and methyltransferases that govern ribosome assembly (pre-assembly). The focus of the paper starts out on pre-assembly thus the sudden connection that the authors try to establish to antibiotic resistance is confusing. A general role on antibiotic tolerance via growth inhibition seems more

likely but the overexpression strain does not inhibit growth. To be clear: If the target of the antibiotic (assembled ribosome) is modified it will promote antibiotic resistance, leading to a clearly elevated minimal inhibitory concentration of the antibiotic in terms of growth inhibition, whereas if the ribosome assembly itself is inhibited it may promote antibiotic tolerance because of the inactivity of the target. Authors do not clearly distinguish between both options and more importantly they do not provide evidence for one or the other option. Antibiotic resistance in terms of minimal inhibitory concentrations with kanamycin is not shown to occur. The growth curves only show a reduced growth rate in the c-di-gmp mutants, they do not show growth inhibition. If the authors want to establish a role in antibiotic resistance they have to demonstrate an impact on the minimal inhibitory concentration. If they want to demonstrate a role on antibiotic tolerance they have to present biphasic killing kinetics that support that role. Given the pleiotropic role of c-di-gmp the impact on growth rate during antibiotic treatment cannot be tied directly to its impact on the interaction with the methyltransferase. Without an established physiological role for this phenomenon of c-di-gmp inhibition of pre-assembly methyltransferases and with these unclear findings authors are not able to establish a mechanism that would explain their observations. The authors would be better off to exclude this part and focus on the physiological mechanism and role of c-di-gmp mediated inhibition of ribosome assembly.

Response: We appreciate your insightful comments. Our previous MIC studies alone did not adequately demonstrate that c-di-GMP promotes resistance. To elucidate the biological function of c-di-GMP in inhibiting ribosome assembly, we conducted bacteria killing assays using kanamycin and ampicillin at 2.5× and 10× MIC concentrations. The results indicate that the mutation at the c-di-GMP binding site K201 on the RlmI protein significantly increases bacterial sensitivity to kanamycin compared with the wild-type strain. Additionally, the overexpression of the c-di-GMP synthase DgcZ induces bacterial tolerance to kanamycin, while the K201 mutation diminishes this effect. Notably, we also observed similar results under ampicillin conditions. These findings suggest that disrupting the interaction between RlmI and c-di-GMP through the K201 mutation can weaken bacterial tolerance to antibiotics.

Based on this evidence, we inferred that c-di-GMP promotes bacterial tolerance to antibiotics by regulating RlmI.

We have updated the relevant sections to reflect these results. (lines 552-564)

Minor concerns:

1. Language throughout the manuscript should be revised. There are a number of grammar mistakes and the manuscript is at times not concise enough, especially towards the end of the manuscript when the impact of antibiotic treatment is explored.

Response: Thank you for pointing this out. We have proofread the manuscript and made necessary modifications, with assistance from International Science Editing.

2. Isolation method of E. coli strains. Seems that this is solely based on growth on

MacConkey Agar which is way too vague. hardly any information on the clinical isolates is provided. A better characterization of the isolates is necessary and should include MICs via Vitek etc. But given the unclear findings it is questionable it does not really make a difference

Response: Thank you for highlighting this issue. In response to these earlier suggestions, we conducted bacteria killing experiments and found that c-di-GMP promotes bacterial tolerance to antibiotics by regulating RImI. Consequently, we have removed the results related to the clinical isolates from this section.

3. how were the strains normalized to quantify ci-di-gmp content - did the authors determine CFUs/ml?

Response: Yes, we utilized absorbance measurements (OD: 600 nm) to calculate the bacterial concentration, which we then used for the quantification of c-di-GMP. We have now included this description in the Methods section. (lines 296-299)

I 381: was controlled is misleading. authors could say has been shown to be controlled by...

Response: For clarity, it has been rephrased as "This increase has been shown to be regulated by the RNA-binding protein CsrA.". (lines 375-376)

I 385: effector? seems right for c-di-gmp not DgcZ

Response: For clarity, it has been rephrased as "DgcZ functions as a synthase that mediates the kanamycin-induced increase in c-di-GMP levels". (lines 380-381)

I389: terminology: replenishment strain?

Response: We thank the reviewer for pointing this out. Regarding the description of the $\Delta dgcZ::dgcZ$ strain, I have revised it to "dgcZ-defective strain complemented with a functional *dgcZ* gene". (lines 384-385)

I510: grammar

Response: This sentence has been rephrased as "we found that the stoichiometry of RImI and its variants with c-di-GMP is not significantly different, each showing a 1:1 binding ratio".

I540: grammar

Response: This term has been rephrased as "The strains were unable to grow at 9 $\mu\text{g}/\text{mL}$ kanamycin, so we determined the growth curves for four other concentrations."

I545: grammar and overstatement

Response: It has been rephrased as "the interaction of c-di-GMP and RImI may enhance antibiotic tolerance".

I549: grammar

Response: We have removed this description of the MIC assay.

I552: grammar

Response: It has been rephrased as “we speculated that c-di-GMP has a promoting effect on kanamycin tolerance.”.

I567: overstatement

Response: We have removed this description of the MIC assay.

I593, I597, I599: overstatement

Response: It has been rephrased as “c-di-GMP is a crucial secondary messenger in prokaryotes, and rRNA methylation occurs in both prokaryotes and eukaryotes. This study revealed that c-di-GMP binds to two rRNA methyltransferases, inhibiting their activities, with RlmI identified as the primary effector of c-di-GMP in ribosome assembly. Molecular dynamics simulations revealed the binding sites and models of c-di-GMP interacting with RlmI. Additionally, killing assays demonstrated that c-di-GMP inhibits ribosome assembly, thereby promoting antibiotic tolerance in E. coli. This research establishes a regulatory pathway linking c-di-GMP to ribosomal functions, underscoring the role of c-di-GMP in antibiotic tolerance.”. (lines 585-594)

I614: what is an upstream regulatory signal and how was this established?

Response: The upstream signals that regulate RlmI methylation function remain unknown, and the biological significance of m5C1962 methylation also requires further exploration.

I631: conflating pre-methyltransferases with post-methyltransferases

Response: In the discussion section, we have removed some of the content regarding rRNA's role in bacterial resistance and added the significance of c-di-GMP's regulation of rRNA methylation in bacterial tolerance. (lines 625-638)

Dear Dr. Xu,

Thank you for the submission of your revised manuscript. Unfortunately, referee 1 was not available to re-review your ms, and I therefore added a new referee, now referee 3. Previous referee 3 is now referee 1 and referee 2 remained the same. All comments and cross-comments are pasted below.

As you will see, both referee 1 and new referee 3 still have several concerns with the revised study, and I think these concerns should be addressed preferably by providing new data. Referee 2 cross-commented on both reports and also agrees that the concerns should be addressed. If you like, we can discuss the revisions beforehand, also in a video chat, if this is helpful.

Please do submit a detailed point-by-point response to all referee concerns with your revised ms.

I look forward to seeing a newly revised form of your manuscript as soon as possible.

Referee #1:

Major concerns:

Figure 5 is still a major problem in this manuscript, Figure 5c should be removed and 5d revised.

Authors try to make a case for antibiotic tolerance but the growth conditions that they use to display it are problematic to say the least:

- The growth curves with subinhibitory concentrations of the antibiotic are done under peculiar conditions, obscuring the exponential growth phase, lag phase and stationary phase. the differences are likely otherwise minimal and probably primarily affect lag phase and stationary phase
- bacteria grown at 25 degrees Celsius
- over 40 hours
- variable scaling of the y -axis, first time point around 10 hours, which is when the exponential phase in untreated cells starts, no replication rates (determined in exponential growth phase)
- Vogel-Bonner Growth medium

- The authors' description of antibiotic tolerance is still not concise enough. The killing kinetics can either show the phenomenon of antibiotic tolerance or antibiotic persistence (slow killing vs biphasic killing) see doi:10.1038/nrmicro.2016.34). it seems that this could actually be antibiotic persistence based on the killing curve. Additional timepoints to really demonstrate a biphasic killing kinetic (including more pronounced plateau-ing) and a determination of the MBK could lead to a clearer assessment. It would also be important to see the killing curve at 100 x of the MIC to really demonstrate antibiotic persistence.

Minor concerns:

- All strains are inhibited in growth at the same minimal inhibitory concentration, thus the term resistance cannot be used in the title of this section
- based on the major concerns authors should revise the terms resistance and tolerance (and persistence) in the text and be particularly mindful when using the term resistance

Referee #2:

Having read manuscript alongside the author's response to the original set of reviews, this version of the manuscript fully addresses the original reviewer's comments.

The only slight point I noticed was the use of 'gram-negative' on lines 84, 87 and elsewhere. The correct use is 'Gram-negative'

and should be changed throughout.

Referee #3:

In this manuscript two 23S rRNA specific methyltransferases (MTases) RlmE and RlmI were identified as targets of second messenger c-di-GMP. c-di-GMP was shown to inhibit methylation activity of both enzymes using in vitro assay. Based on this observation authors assume that this second messenger modulates ribosome assembly. However, no evidence for ribosome assembly defect at different level of c-di-GMP was observed unless kanamycin was added. It is proposed that inhibition of RlmI methylase activity by c-di-GMP leads directly to ribosome assembly defect, which is observed only in the presence of antibiotic kanamycin.

Depletion of RlmI does not cause ribosome assembly defects in the absence of antibiotics according to this work and an earlier publication (PMID: 32174967). Kanamycin induced accumulation of 45S assembly defective ribosomal particles is identical in the WT and in the Δ RlmI strain (Fig. 2F). Thus, there is no evidence for involvement of RlmI in the ribosome LSU assembly either in the presence or in the absence of kanamycin.

The manuscript is written in a confusing style, which makes it difficult to understand. For example:

Line 426 „With kanamycin stimulation, the elevated c-di-GMP in WT cells inhibited 427 RlmI, resulting in missing rRNA methylation”

Did I miss the important data. I cannot see any evidence for inhibition of RlmI activity (methylation of C1962) in either WT or mutant cells. This is very important point.

Based on the results of this manuscript and the data published earlier, the methylation of C1962 is not needed for ribosome LSU assembly either in the presence or absence of kanamycin. Instead, kanamycin inhibits synthesis of proteins including ribosomal proteins. It is known for other antibiotics (chloramphenicol and erythromycin) that they cause nonstoichiometric expression of r-proteins, which leads to accumulation of 45S-like particles (PMID: 1764521; 21320180). Therefore, kanamycin dependent ribosome assembly defect is most likely due to the inhibition of ribosomal protein production and not due to inhibition of RlmI.

Additional experiments needed:

Does elevated level of c-di-GMP affect 23S rRNA methylation by RlmI in vivo?

Is there any difference of methylation level between 23S rRNA of 45S assembly deficient particles and 70S ribosomes? If yes, one could conclude that synthesis of m5C1962 is indeed important for ribosome LSU maturation to the 50S ribosome. If not, the accumulation of 45S particles is not due to the inhibition of RlmI methylase activity.

Does addition of kanamycin lead to non-stoichiometric production of ribosomal proteins in bacteria as it was shown for chloramphenicol and erythromycin (if it is too much, this point must be at least discussed).

Minor points

In vitro methylation assay needs more detailed documentation. The RNA substrates must be specified also in the methods section. It is surprising to find that RlmE methylates 23S rRNA fragment. Earlier publication has demonstrated that methylation of in vitro transcribed 23S rRNA by RlmE is very inefficient (around 1 %) (PMID: 10983982). In contrast, RlmI can easily methylate protein-free 23S rRNA (PMID: 18786544). In the Fig. 1 B - D both enzymes RlmE and RlmI are apparently equally efficient. Methylation was quenched by heating samples at 95 {degree sign}C. This treatment can lead to RNA degradation. It should be shown that the second peak on the HPLC chromatogram is indeed methylated 23S rRNA fragment (either mass-spectrometry or nucleoside analysis). It seems unlikely that single methyl group leads to such a big difference in elution time. At least unmethylated control RNA samples need to be analyzed in the same way as a control.

In description of sucrose density gradient analysis, it reads: "The suspension was recovered to 25 components and quantified by measuring absorbance at 260 nm using a NanoDrop 2000 spectrophotometer". This method is expected to result in 25 columns representing the 25 fractions. Instead, in the Fig 2 (A, B, and F) smooth curves of optical density are shown. It appears that the optical profile has been averaged over the fractions to get smooth lines. This approach can potentially lead to misinterpretation of the ribosome peak shapes, which is important for interpreting ribosome assembly intermediate particles. In particular in quantitative terms (e.g. Fig. 2B).

Cross-comments from referee 2 :

Having read through reviewer 1's concerns, I think they look pretty reasonable. Given the simplicity of most antimicrobial assays, it shouldn't be a major issue for the authors to repeat these assays to address the specific issues of the reviewer. Otherwise, the text could be edited reasonably easily to mostly address these points with only a little additional work needed.

My thoughts on reviewer 3's points are as follows:

• Additional experiments needed:

Does elevated level of c-di-GMP affect 23S rRNA methylation by RlmI in vivo?

The issue with doing anything in vivo with elevated cdG is you get all sorts of confounding inputs from other systems changing

their regulation. While this would be good to see, I think the in vitro evidence presented by the authors for RlmI RNA methylation, and the inhibition of this by cdG binding is already pretty good.

- Is there any difference of methylation level between 23S rRNA of 45S assembly deficient particles and 70S ribosomes? If yes, one could conclude that synthesis of m⁵C1962 is indeed important for ribosome LSU maturation to the 50S ribosome. If not, the accumulation of 45S particles is not due to the inhibition of RlmI methylase activity.
- Does addition of kanamycin lead to non-stoichiometric production of ribosomal proteins in bacteria as it was shown for chloramphenicol and erythromycin (if it is too much, this point must be at least discussed).

These are both fair points. My view was that the authors had strongly demonstrated that: a) RlmI binds cdG; and b) RlmI methylates rRNA and this is inhibited by cdG binding. This pair of findings together allow them to make a plausible explanation for the antimicrobial tolerance/persistence phenomena observed with the various different mutants tested. While for me this was enough, I think it would be reasonable to ask for these experiments to be done - it would certainly make for a stronger story.

Referee #1:

Major concerns:

Figure 5 is still a major problem in this manuscript, Figure 5c should be removed and 5d revised.

Authors try to make a case for antibiotic tolerance but the growth conditions that they use to display it are problematic to say the least:

- The growth curves with subinhibitory concentrations of the antibiotic are done under peculiar conditions, obscuring the exponential growth phase, lag phase and stationary phase. the differences are likely otherwise minimal and probably primarily affect lag phase and stationary phase
- bacteria grown at 25 degrees Celsius
- over 40 hours
- variable scaling of the y -axi, first time point around 10 hours, which is when the exponential phase in untreated cells starts, no replication rates (determined in exponential growth phase)
- Vogel-Bonner Growth medium

Response: Thank you for your comments. We appreciate your suggestion regarding the use of growth curves. We agree that relying solely on growth curves may not provide accurate conclusions. As per your recommendation, we have removed this section from the manuscript.

- The authors' description of antibiotic tolerance is still not concise enough. The killing kinetics can either show the phenomenon of antibiotic tolerance or antibiotic persistence (slow killing vs biphasic killing) see doi:10.1038/nrmicro.2016.34). it seems that this could actually be antibiotic persistence based on the killing curve. Additional timepoints to really demonstrate a biphasic killing kintec (including more pronoundc plateau-ing) and a determination of the MBK could lead to a clearer assessment. It would also be important to see the killing curve at 100 x of the MIC to really demonstrate antibiotic persistence.

Response: Thank you for your insightful and helpful comments. We did not sufficiently distinguish between antibiotic tolerance and antibiotic persistence in our original manuscript. Following your suggestion, we have now included data for antibiotic concentrations of 20x and 100x and extended the antibiotic exposure time. Additionally, we have calculated the MDK99 and MDK99.99 values as recommended. The results revealed that the MDK99 for the *rlmI*^{K201A} strain was similar to that of the WT strain, indicating that the loss of c-di-GMP regulation on RlmI does not significantly affect general antibiotic tolerance. However, the MDK99.99 for *rlmI*^{K201A} was significantly lower and remained unaffected by antibiotic concentration, underscoring the role of c-di-GMP regulation of RlmI in enhancing bacterial persistence under antibiotic stress (Fig. 5D).

We appreciate your careful review and believe that these revisions have strengthened the clarity and accuracy of our findings.

We have updated the relevant sections to reflect these results. (lines 565-587)

Minor concerns:

- All strains are inhibited in growth at the same minimal inhibitory concentration, thus the term resistance cannot be used in the title of this section

Response: We have revised the manuscript as suggested.

- based on the major concerns authors should revise the terms resistance and tolerance (and persistence) in the text and be particularly mindful when using the term resistance

Response: We have revised the manuscript as suggested.

Referee #2:

Having read manuscript alongside the author's response to the original set of reviews, this version of the manuscript fully addresses the original reviewer's comments.

The only slight point I noticed was the use of 'gram-negative' on lines 84, 87 and elsewhere. The correct use is 'Gram-negative' and should be changed throughout.

Response: Thank you for your positive feedback and for recognizing the improvements in our revised manuscript. In response to your suggestion, we have made the requested revision to Gram-negative.

Referee #3:

In this manuscript two 23S rRNA specific methyltransferases (MTases) RlmE and RlmI were identified as targets of second messenger c-di-GMP. c-di-GMP was shown to inhibit methylation activity of both enzymes using in vitro assay. Based on this observation authors assume that this second messenger modulates ribosome assembly. However, no evidence for ribosome assembly defect at different level of c-di-GMP was observed unless kanamycin was added. It is proposed that inhibition of RlmI methylase activity by c-di-GMP leads directly to ribosome assembly defect, which is observed only in the presence of antibiotic kanamycin.

Depletion of RlmI does not cause ribosome assembly defects in the absence of antibiotics according to this work and an earlier publication (PMID: 32174967). Kanamycin induced accumulation of 45S assembly defective ribosomal particles is identical in the WT and in the Δ RlmI strain (Fig. 2F). Thus, there is no evidence for involvement of RlmI in the ribosome LSU assembly either in the presence or in the absence of kanamycin.

Did I miss the important data. I cannot see any evidence for inhibition of RlmI activity (methylation of C1962) in either WT or mutant cells. This is very important point.

Based on the results of this manuscript and the data published earlier, the methylation of C1962 is not needed for ribosome LSU assembly either in the presence or absence of kanamycin. Instead, kanamycin inhibits synthesis of proteins including ribosomal proteins. It is known for other antibiotics (chloramphenicol and erythromycin) that they cause non-stoichiometric expression of r-proteins, which leads to accumulation of 45S-like particles (PMID: 1764521; 21320180). Therefore, kanamycin dependent ribosome assembly defect is most likely due to the inhibition of ribosomal protein production and not due to inhibition of RlmI.

Response: Thank you for your valuable comments and thoughtful suggestions. We fully agree with your point that, as an important aspect of the manuscript, it is essential to demonstrate that c-di-GMP can influence the methylation of the C1962 site in vivo through RlmI. Additionally, further evidence is needed to show that the methylation of the C1962 site is crucial for ribosome assembly. In response to your suggestion, we have conducted additional experiments and have updated the manuscript to include these new results and their descriptions.

Specific details of the changes can be found in the following responses.

Additional experiments needed:

Does elevated level of c-di-GMP affect 23S rRNA methylation by RlmI in vivo?

Response: Thank you for your constructive comments. In the previous version of the manuscript, we had already examined how c-di-GMP affects 23S rRNA methylation by RlmI in vivo. However, we apologize for not providing a more detailed explanation of this experiment. We have now included the following additional information in the Results section: To increase the concentration of c-di-GMP, we constructed an

overexpression strain of DcgZ ($\Delta dgcZ dgcZ^+$) as shown in Figure 5A. The enzymatic activity of RlmI was assessed by measuring the endogenous C1962 methylation level. Our results showed that, compared to the $\Delta dgcZ$ and wild-type strains, the $\Delta dgcZ dgcZ^+$ strain exhibited a significantly reduced C1962 methylation level, indicating that RlmI activity was suppressed in this strain (Fig. 5B). These findings provide in vivo evidence that c-di-GMP regulates 23S rRNA methylation through RlmI.

We hope that this additional data and explanation address your concerns and further strengthen the manuscript.

We have updated the relevant sections to reflect these results. (lines 560-564)

Is there any difference of methylation level between 23S rRNA of 45S assembly deficient particles and 70S ribosomes? If yes, one could conclude that synthesis of m5C1962 is indeed important for ribosome LSU maturation to the 50S ribosome. If not, the accumulation of 45S particles is not due to the inhibition of RlmI methylase activity.

Response: Thank you for your constructive comments. To investigate the role of methylation at the C1962 site in ribosome assembly, we isolated and analyzed the relative abundance of methylated C1962 in both the 45S ribosomal precursor and the 50S ribosomal large subunit, using 26S rRNA as a baseline reference. The results showed that methylated C1962 was more abundant in the 50S ribosomal large subunit, suggesting that methylation at C1962 may play an important functional role in the assembly of the ribosomal large subunit (Fig. 2G).

We have updated the relevant sections to reflect these results. (lines 450-456)

Does addition of kanamycin lead to non-stoichiometric production of ribosomal proteins in bacteria as it was shown for chloramphenicol and erythromycin (if it is too much, this point must be at least discussed).

Response: Thank you for your insightful comments. Indeed, the current manuscript does not address the role of kanamycin in ribosome assembly, which is also a question that we are very interested in exploring. To answer this, we plan to identify the components of the 45S ribosome intermediates and analyze their structure in our future work. We have now included a statement about this in the Discussion section of the manuscript to clarify our plans.

We have updated the relevant sections in Discussion. (lines 648-656)

Minor points

In vitro methylation assay needs more detailed documentation. The RNA substrates must be specified also in the methods section. It is surprising to find that RlmE methylates 23S rRNA fragment. Earlier publication has demonstrated that methylation of in vitro transcribed 23S rRNA by RlmE is very inefficient (around 1 %) (PMID: 10983982). In contrast, RlmI can easily methylate protein-free 23S rRNA (PMID: 18786544). In the Fig. 1 B - D both enzymes RlmE and RlmI are apparently equally efficient. Methylation was quenched by heating samples at 95 {degree sign}C. This treatment can lead to RNA degradation. It should be shown that the

second peak on the HPLC chromatogram is indeed methylated 23S rRNA fragment (either mass-spectrometry or nucleoside analysis). It seems unlikely that single methyl group leads to such a big difference in elution time. At least unmethylated control RNA samples need to be analyzed in the same way as a control.

Response: As you pointed out, the methyltransferase activity of RlmE and RlmI is relatively weak in the in vitro assays. We have optimized the purification and storage conditions multiple times, but unfortunately, we were only able to achieve approximately 5% catalytic efficiency. We have now included more details of these experimental conditions in the Methods section of the manuscript to provide better transparency and clarity. (lines 133-151)

Regarding the HPLC analysis, we did include standard samples as negative controls. We have now added the relevant spectra to Figure S1. These results show that, after heating at 95°C, the unmethylated substrate RNA did not decompose into any additional impurities. Additionally, the methylated rRNA standard was used as a reference to validate the synthetic product.

In description of sucrose density gradient analysis, it reads: "The suspension was recovered to 25 components and quantified by measuring absorbance at 260 nm using a NanoDrop 2000 spectrophotometer". This method is expected to result in 25 columns representing the 25 fractions. Instead, in the Fig 2 (A, B, and F) smooth curves of optical density are shown. It appears that the optical profile has been averaged over the fractions to get smooth lines. This approach can potentially lead to misinterpretation of the ribosome peak shapes, which is important for interpreting ribosome assembly intermediate particles. In particular in quantitative terms (e.g. Fig. 2B).

Response: Thank you for your suggestions. We appreciate your recommendation to use bar graphs for presenting the data, as this approach indeed provides a clearer and more intuitive way to display the results. However, to better illustrate the ribosomal subunits (large subunit, small subunit, and assembly intermediates), we have chosen to fit the data. This fitting approach is commonly used in the field for this type of analysis, and we believe it provides a more accurate representation of the dynamics.

Regarding the quantitative issue, in Figure 2B, under the condition of RlmI overexpression, we clearly observed that the 45S assembly precursor nearly disappeared, which suggests that RlmI can inhibit the generation of 45S ribosomal precursors. On the other hand, the 45S and 50S ribosomal particles do not fully separate in the density gradient centrifugation, so calculating the 45S ribosome content based on peak areas or bar chart areas may not be entirely accurate.

Thank you again for raising this important question. To provide readers with more detailed information, we have uploaded the absorbance data from this experiment to the journal.

We hope these clarifications help address your concerns and improve the manuscript. Thank you once again for your constructive feedback.

The manuscript is written in a confusing style, which makes it difficult to understand.

For example:

**Line 426 „With kanamycin stimulation, the elevated c-di-GMP in WT cells inhibited
427 RlmI, resulting in missing rRNA methylation"**

Response: Thank you for your valuable feedback. We apologize for the confusion caused by the manuscript's writing style. We have restructured certain sections and clarified the language to make the text more accessible and easier to follow. Additionally, we have asked a native English speaker to proofread and refine the manuscript.

Thank you again for your constructive criticism.

Dear Dr. Xu,

Thank you for the submission of your revised manuscript. We have now received the enclosed reports from the referees who were asked to assess it.

As you will see, both referees still have important concerns and all will need to be addressed in order to proceed with your ms for publication here. Please revise the ms as we discussed and send us a final version as soon as possible.

Please note that referee 3 suggests to add "in the presence of Kanamycin" to the title of Figure 1.

A few editorial requests will also need to be addressed:

- "Highlights" need to be removed from the ms file.
- Materials & Methods should be Methods.
- Data and software availability should be renamed to "Data Availability Section" and this section needs to be moved to the end of the Methods.
- The Author Contributions need to be removed from the ms file. All contributions are entered during online ms submission.
- The manuscript sections should be in the following order: Title page - Abstract & Keywords - Introduction - Results - Discussion - Methods - Data Availability - Acknowledgments - Disclosure Statement & Competing Interests - References - Figure Legends
- References: need to be alphabetical, not numerical; et al needs to be used after 10 author names; DOIs should only be used for preprints and datasets that have not been published yet. The EMBO reports reference style is also in EndNote.
- All Figures need to be provided as individual production quality figure files as .eps, .tif, .jpg (one file per figure).
- Appendix file: "S" is missing in the figure titles throughout the file - e.g. Appendix Figure 1 should be Appendix Figure S1, etc.
- Movies: the source file names, titles and callouts in the ms need correction to Movie EV1, Movie EV2; we need a legend for each movie in a readme.txt file so that each legend can be zipped together with its movie and upldd as one zip folder - Movie EV1 and Movie EV2.
- The Reagents and Tools table needs to be provided as a separate Word file
- All Source data need to be uploaded separately as 1 file per figure.
- Please indicate the statistical test used for data analysis in the legend of figure 1A
- Please note that information related to n is missing in the legend of figure 5C.
- Although 'n' is provided, please describe the nature of entity for 'n' in the legends of figures 1D; 2C-E; 4D, F; 5A, B, D; 6C.
- Please note that the error bars are not defined in the legend of figure 5C.

I would like to suggest some changes to the ms title and abstract. Please let me know whether you agree with the following:

c-di-GMP inhibits rRNA Methylation and impairs Ribosome Assembly in the presence of Kanamycin

Cyclic diguanosine monophosphate (c-di-GMP) is a ubiquitous bacterial secondary messenger with diverse functions. A previous *Escherichia coli* proteome microarray identified that c-di-GMP binds to the 23S rRNA methyltransferases RlmI and RlmE. Here we show that c-di-GMP inhibits RlmI activity in rRNA methylation assays, and that it modulates ribosome assembly in the presence of kanamycin. Molecular dynamics simulation and mutagenesis studies reveal that c-di-GMP binds to RlmI at residues R64, R103, G114, and K201. Structural simulations indicate that c-di-GMP quenches RlmI activity by inducing the closure of the catalytic pocket. We also show that c-di-GMP promotes antibiotic persistence through RlmI. Binding and methylation assays indicate that the inhibitory effect of c-di-GMP on RlmI is conserved across various pathogenic bacteria. Our data suggest an unexpected role for c-di-GMP in regulating ribosome assembly under stress through the inhibition of rRNA methyltransferases.

EMBO press papers are accompanied online by A) a short (1-2 sentences) summary of the findings and their significance, B) 2-3 bullet points highlighting key results and C) a synopsis image that is exactly 550 pixels wide and 200-600 pixels high (the height is variable). The synopsis image should provide a sketch of the major findings, like a graphical abstract. Please note that text needs to be readable at the final size. Please send us this information along with the final manuscript.

I look forward to seeing a final version of your manuscript when it is ready.

Yours sincerely,

Referee #1:

Reviewers have addressed my concerns and have implemented MDK which has significantly improved Figure 5 and the associated text. The authors' conclusion is that RlmI promotes antibiotic persistence. The curve that the authors show however does not display the hallmarks of persistence (rapid initial killing followed by slow killing which complicates the distinction to antibiotic tolerance. The kill curve shows slow initial killing followed by faster killing followed by slower killing. In the absence of initial fast killing followed by slow killing it is hard to state that this is persistence. Tolerance seems more appropriate since the killing dynamics seem similar at the time of MDK99,99. But it is hard to make a decision without a clean killing curve. This could be due to the usage of Vogel Bonner medium which I still find inappropriate. Authors should repeat the killing kinetic with LB medium in the entire experiment instead of growing the bacteria in LB medium overnight and then switching to a different medium. That, in fact may have affected the killing kinetic. Authors should also state the MDK values.

Referee #3:

This paper has certainly a value in the c-di-GMP related experiments. However, the conclusions about the ribosome assembly inhibition are premature and not supported by the experiments. My comments:
The authors have made additional experiments and revised the manuscript. However, the main point concerning the regulatory role of c-di-GMP during ribosome assembly remains elusive. The conclusion: "This study revealed an unexpected functional role of
46 c-di-GMP in regulating ribosome assembly through the inhibition of
47 rRNA methyltransferases, highlighting an unexpected but crucial member
48 among the c-di-GMP effectors" is not supported by the results. First, inhibition of ribosome assembly appears only in the presence of kanamycin, an aminoglycoside antibiotic. Aminoglycosides inhibit ribosome assembly (see PMID: 14570276 PMID: 11959595). The only evidence for ribosome assembly in this manuscript is an accumulation of 45S-like particles in the *del-rlmI* strain in the presence of kanamycin. As the authors refused to demonstrate the measured ribosome sucrose density gradient profiles, it is impossible to differentiate between the result and the artwork.

Response to referees/editors

Please note that referee 3 suggests to add "in the presence of Kanamycin" to the title of Figure 1.

Response: We have revised the manuscript in accordance with the suggestion.

A few editorial requests will also need to be addressed:

- "Highlights" need to be removed from the ms file.

Response: We have revised the manuscript in accordance with the suggestion.

- Materials & Methods should be Methods.

Response: We have revised the manuscript in accordance with the suggestion.

- Data and software availability should be renamed to "Data Availability Section" and this section needs to be moved to the end of the Methods.

Response: We have revised the manuscript in accordance with the suggestion.

- The Author Contributions need to be removed from the ms file. All contributions are entered during online ms submission.

Response: We have revised the manuscript in accordance with the suggestion.

- The manuscript sections should be in the following order: Title page - Abstract & Keywords - Introduction - Results - Discussion - Methods - Data Availability - Acknowledgments - Disclosure Statement & Competing Interests - References - Figure Legends

Response: We have revised the manuscript in accordance with the suggestion.

- References: need to be alphabetical, not numerical; et al needs to be used after 10 author names; DOIs should only be used for preprints and datasets that have not been published yet. The EMBO reports reference style is also in EndNote.

Response: We have revised the References in accordance with the suggestion.

- All Figures need to be provided as individual production quality figure files as .eps, .tif, .jpg (one file per figure).

Response: We have revised the Figures in accordance with the suggestion.

- Appendix file: "S" is missing in the figure titles throughout the file - e.g. Appendix Figure 1 should be Appendix Figure S1, etc.

Response: We have revised the manuscript in accordance with the suggestion.

- Movies: the source file names, titles and callouts in the ms need correction to Movie EV1, Movie EV2; we need a legend for each movie in a readme txt file so that each legend can be zipped together with its movie and uplidd as one zip folder - Movie EV1 and Movie EV2.

Response: We have revised the manuscript in accordance with the suggestion.

- The needs to be provided as a separate Word file

Response: We have added the Reagents and Tools table in accordance with the suggestion.

- All Source data need to be uploaded separately as 1 file per figure.

Response: We have separated the Source data in accordance with the suggestion.

- Please indicate the statistical test used for data analysis in the legend of figure 1A

Response: Initially, we employed a t-test to evaluate the differences in signal between the two proteins. However, we recognized that each condition in our experiment had only two replicates. Given this limited sample size, the statistical power of the t-test was insufficient to produce reliable p-values. Small sample sizes can lead to unreliable and potentially misleading statistical inferences, which was a concern in our analysis.

Considering these limitations, we decided to omit the p-values from our results. Despite the absence of statistical significance values, the differences in signal between the c-di-GMP and the control groups are markedly distinct and clearly observable in the data. The visual separation of these signals supports the conclusion that the observed differences are meaningful.

- Please note that information related to n is missing in the legend of figure 5C.

Response: We have revised the manuscript in accordance with the suggestion.

- Although 'n' is provided, please describe the nature of entity for 'n' in the legends of figures 1D; 2C-E; 4D, F; 5A, B, D; 6C.

Response: We have added the information in legend.

- Please note that the error bars are not defined in the legend of figure 5C.

Response: We have added the information in legend.

I would like to suggest some changes to the ms title and abstract. Please let me know whether you agree with the following:

c-di-GMP inhibits rRNA Methylation and impairs Ribosome Assembly in the presence of Kanamycin

Cyclic diguanosine monophosphate (c-di-GMP) is a ubiquitous bacterial secondary messenger with diverse functions. A previous *Escherichia coli* proteome microarray identified that c-di-GMP binds to the 23S rRNA methyltransferases RlmI and RlmE. Here we show that c-di-GMP inhibits RlmI activity in rRNA methylation assays, and that it modulates ribosome assembly in the presence of kanamycin. Molecular dynamics simulation and mutagenesis studies reveal that c-di-GMP binds to RlmI at residues R64, R103, G114, and K201. Structural simulations indicate that c-di-GMP quenches RlmI activity by inducing the closure of the catalytic pocket. We also show that c-di-GMP promotes antibiotic persistence through RlmI. Binding and methylation assays indicate that the inhibitory effect of c-di-GMP on RlmI is conserved across various pathogenic bacteria. Our data suggest an unexpected role for c-di-GMP in regulating ribosome assembly under stress through the inhibition of rRNA methyltransferases.

Response: We agree that the revised title and abstract more accurately

encapsulates the dual role of c-di-GMP in both rRNA methylation and ribosome assembly.

Referee #1:

Reviewers have addressed my concerns and have implemented MDK which has significantly improved Figure 5 and the associated text. The authors' conclusion is that RImI promotes antibiotic persistence. The curve that the authors show however does not display the hallmarks of persistence (rapid initial killing followed by slow killing which complicates the distinction to antibiotic tolerance. The kill curve shows slow initial killing followed by faster killing followed by slower killing. In the absence of initial fast killing followed by slow killing it is hard to state that this is persistence. Tolerance seems more appropriate since the killing dynamics seem similar at the time of MDK99,99. But it is hard to make a decision without a clean killing curve. This could be due to the usage of Vogel Bonner medium which I still find inappropriate. Authors should repeat the killing kinetic with LB medium in the entire experiment instead of growing the bacteria in LB medium overnight and then switching to a different medium. That, in fact may have affected the killing kinetic. Authors should also state the MDK values.

Response: We would like to extend our sincere gratitude for your insightful comments on our manuscript. In our initial submission, we reported that the *rlmI*^{K201A} strain exhibited no significant change in MDK₉₉ compared to the WT strain, while MDK_{99,99} was significantly reduced. Based on these findings, we concluded that c-di-GMP promotes bacterial persistence in *E. coli*. However, we acknowledge that the observed slow decline in the kill curves was inconsistent with the expected persistence phenotype of *E. coli*.

As you correctly pointed out, the slow initial decline in the kill curves suggested a discrepancy with the persistence phenotype. Additionally, the practice of replacing the medium during the killing assay might have introduced other endogenous stress responses in the bacteria, potentially confounding the results. To address these concerns and obtain more convincing data, we conducted the kill curve experiments entirely in LB medium without medium replacement throughout the assay. The revised experiments demonstrated an early rapid decline in bacterial viability, aligning with previous studies on *E. coli* tolerance or persistence.

Furthermore, we re-analyzed the MDK₉₉ and MDK_{99,99} values for three strains: WT, *rlmI*^{K201A}, and Δ *dgcZ* strain. The results showed that both the *rlmI*^{K201A} and Δ *dgcZ* strains exhibited significant reductions in MDK₉₉ and MDK_{99,99} compared to the WT. These findings showed that

the c-di-GMP regulation of the RlmI pathway plays a crucial role in modulating bacterial antibiotic tolerance.

Based on the newly obtained data, we have revised the manuscript to reflect that c-di-GMP enhances antibiotic tolerance in *E. coli* by regulating the RlmI pathway. We have updated **Figure 6 C-D** and the corresponding sections in the Results and Discussion to incorporate these new findings.

Figure 6. c-di-GMP regulates ribosome assembly to promote antibiotic tolerance.

Referee #3

This paper has certainly a value in the c-di-GMP related experiments. However, the conclusions about the ribosome assembly inhibition are premature and not supported by the experiments. My comments:

The authors have made additional experiments and revised the manuscript. However, the main point concerning the regulatory role of c-di-GMP during ribosome assembly remains elusive. The conclusion: "This study revealed an unexpected functional role of c-di-GMP in regulating ribosome assembly through the inhibition of rRNA methyltransferases, highlighting an unexpected but crucial member among the c-di-GMP effectors" is not supported by the results. First, inhibition of ribosome assembly appears only in the presence of kanamycin, an aminoglycoside antibiotic. Aminoglycosides inhibit ribosome assembly (see PMID: 14570276 PMID: 11959595). The only evidence for ribosome assembly in this manuscript is an accumulation of 45S-like particles in the del-rlmI strain in the presence of kanamycin. As the authors refused to demonstrate the measured ribosome sucrose density gradient profiles, it is impossible to differentiate between the result and the artwork.

Response: We sincerely thank you for your suggestions. We appreciate your recommendation to explore the impact of varying Mg^{2+} concentrations on ribosome profiles. Initially, we conducted ribosome density gradient centrifugation experiments at 10 mM Mg^{2+} . Under these

conditions, we observed a reduction in mature 70S ribosomes and an increase in 50S subunits; however, signals corresponding to the 45S ribosomal subunits were not detected. To further investigate this, we performed additional experiments at lower Mg^{2+} concentrations and found that at 0.5 mM Mg^{2+} , a distinct 45S subunit signal became apparent. This observation aligns with previous studies, specifically Arai *et al.* (PNAS, 2015, PMID: 26261349), supporting the presence of immature ribosomal subunits under reduced Mg^{2+} conditions.

To provide a more comprehensive dataset, we have included additional ribosome density gradient centrifugation experiments at 10 mM Mg^{2+} . The results demonstrate that elevated c-di-GMP concentrations suppress the formation of 70S ribosomes. Importantly, mutations R64 or R103 of RlmI effectively prevent this suppression, indicating their role in ribosome assembly (**Figure 3A**). Furthermore, we analyzed the methylation levels of C1962 in both 70S and 50S ribosomal subunits. Our findings reveal that in strains with high c-di-GMP expression, the C1962 methylation level in the 50S subunit is significantly reduced (**Figure 3B**). This suggests that the 50S subunits in these strains likely contain immature ribosomes, providing additional evidence for the impact of c-di-GMP on ribosome maturation.

In summary, by presenting ribosome density gradient centrifugation data at both 0.5 mM and 10 mM Mg^{2+} , we are able to observe both the alterations in mature ribosome populations and the presence of 45S

ribosomal precursors. This dual approach not only corroborates our findings with existing literature but also enriches the overall robustness of our results, offering a more complete understanding of ribosome maturation under varying Mg^{2+} concentrations.

Figure 3. Ribosome assembly and C1962 methylation in response to varying Mg^{2+} concentrations and c-di-GMP levels in the presence of kanamycin.

Zhaowei Xu
Fujian medical university
China

Dear Dr. Xu,

I am very pleased to accept your manuscript for publication in the next available issue of EMBO reports. Thank you for your contribution to our journal.

Some of the text in the synopsis image is very small, and the methylation tag can hardly be read. It would be very helpful if you could send us a new synopsis image with larger font size. Thank you.

Yours sincerely,

Referee #1:

The authors have addressed my concerns. However, I have to say, the difference in tolerance is really small and the graph depicting tolerance is still not clear and obvious enough to show a different slope, probably because the difference is very small. Authors now explicitly state the MBK values though so the reader can decide if this is significant or not.
